# Neural Estimation of Submodular Functions with Applications to Differentiable Subset Selection

**Abir De      Soumen Chakrabarti**
Indian Institute of Technology Bombay
{abir,soumen}@cse.iitb.ac.in

## Abstract

Submodular functions and variants, through their ability to characterize diversity and coverage, have emerged as a key tool for data selection and summarization. Many recent approaches to learn submodular functions suffer from limited expressiveness. In this work, we propose FLEXSUBNET, a family of flexible neural models for both monotone and non-monotone submodular functions. To fit a latent submodular function from (set, value) observations, FLEXSUBNET applies a concave function on modular functions in a recursive manner. We do not draw the concave function from a restricted family, but rather learn from data using a highly expressive neural network that implements a differentiable quadrature procedure. Such an expressive neural model for concave functions may be of independent interest. Next, we extend this setup to provide a novel characterization of monotone $\alpha$-submodular functions, a recently introduced notion of approximate submodular functions. We then use this characterization to design a novel neural model for such functions. Finally, we consider learning submodular set functions under distant supervision in the form of (perimeter-set, high-value-subset) pairs. This yields a novel subset selection method based on an order-invariant, yet greedy sampler built around the above neural set functions. Our experiments on synthetic and real data show that FLEXSUBNET outperforms several baselines.

## 1    Introduction

Owing to their strong characterization of diversity and coverage, submodular functions and their extensions, *viz.*, weak and approximate submodular functions, have emerged as a powerful machinery in data selection tasks [46, 84, 35, 66, 91, 63, 5, 23, 75]. We propose trainable parameterized families of submodular functions under two supervision regimes. In the first setting, the goal is to estimate the submodular function based on (set, value) pairs, where the function outputs the value of an input set. This problem is hard in the worst case for poly$(n)$ value oracle queries [31]. This has applications in auction design where one may like to learn a player's valuation function based on her bids [6]. In the second setting, the task is to learn the submodular function under the supervision of (perimeter-set, high-value-subset) pairs, where high-value-subset potentially maximizes the underlying function against all other subsets of perimeter-set. The trained function is expected to extract high-value subsets from freshly-specified perimeter sets. This scenario has applications in itemset prediction in recommendation [78, 79], data summarization [50, 2, 4, 10, 74], *etc*.

### 1.1    Our contributions

Driven by the above motivations, we propose (i) a novel family of highly expressive neural models for submodular and $\alpha$-submodular functions which can be estimated under the supervisions of both (set, value) and (perimeter-set, high-value-subset) pairs; (ii) a novel permutation adversarial training method for differentiable subset selection, which efficiently trains submodular and $\alpha$-submodular functions based on (perimeter-set, high-value-subset) pairs. We provide more details below.

36th Conference on Neural Information Processing Systems (NeurIPS 2022).

**Neural models for submodular functions.** We design FLEXSUBNET, a family of flexible neural models for monotone, non-monotone submodular functions and monotone $\alpha$-submodular functions.

— *Monotone submodular functions.* We model a monotone submodular function using a recursive neural model which outputs a concave composed submodular function [27, 76, 52] at every step of recursion. Specifically, it first computes a linear combination of the submodular function computed in the previous step and a modular function and, then applies a concave function to the result to output the next submodular function.

— *Monotone $\alpha$-submodular function.* Our proposed model for submodular function rests on the principle that a concave composed submodular function is always submodular. However, to the best of our knowledge, there is no known result for an $\alpha$-submodular function. We address this gap by showing that an $\alpha$-submodular function can be represented by applying a mapping $\varphi$ on a positive modular set function, where $\varphi$ satisfies a second-order differential inequality. Subsequently, we provide a neural model representing the universal approximator of $\varphi$, which in turn is used for modeling an $\alpha$-submodular function.

— *Non-monotone submodular functions.* By applying a non-monotone concave function on modular function, we extend our model to non-monotone submodular functions.

Several recent models learn subclasses of submodular functions [53, 77, 79]. Bilmes and Bai [7] present a thorough theoretical characterization of the benefits of network depth with fixed concave functions, in a general framework called deep submodular functions (DSF). DSF leaves open all design choices: the number of layers, their widths, the DAG topology, and the choice of concave functions. All-to-all attention has replaced domain-driven topology design in much of NLP [18]. Set transformers [51] would therefore be a natural alternative to compare against DSF, but need more memory and computation. Here we explore the following third, somewhat extreme trade-off: we restrict the topology to a single recursive chain, thus providing an effectively plug-and-play model with no topology and minimal hyperparameter choices (mainly the length of the chain). However, we compensate with a more expressive, trainable concave function that is shared across all nodes of the chain. Our experiments show that our strategy improves ease of training and predictive accuracy beyond both set transformers and various DSF instantiations with fixed concave functions.

**Permutation insensitive differentiable subset selection.** It is common [77, 70, 63] to select a subset from a given dataset by sequentially sampling elements using a softmax distribution obtained from the outputs of a set function on various sets of elements. At the time of learning set functions based on (perimeter-set, high-value-subset) pairs, this protocol naturally results in order-sensitivity in the training process for learning set functions. To mitigate this, we propose a novel max-min optimization. Such a formulation sets forth a game between an adversarial permutation generator and the set function learner — where the former generates the worst-case permutations to induce minimum likelihood of training subsets and the latter keeps maximizing the likelihood function until the estimated parameters become permutation-agnostic. To this end, we use a Gumbel-Sinkhorn neural network [57, 68, 17, 69] as a neural surrogate of hard permutations, that expedites the underlying training process and allows us to avoid combinatorial search on large permutation spaces.

**Experiments.** We first experiment with several submodular set functions and synthetically generated examples, which show that FLEXSUBNET recovers the function more accurately than several baselines. Later experiments with several real datasets on product recommendation reveal that FLEXSUBNET can predict the purchased items more effectively and efficiently than several baselines.

## 2 Related work

**Deep set functions.** Recent years have witnessed a surge of interest in deep learning of set functions. Zaheer et al. [86] showed that any set function can be modeled using a symmteric aggregator on the feature vectors associated with the underlying set members. Lee et al. [51] proposed a transformer based neural architecture to model set functions. However, their work do not focus on modeling or learning submodular functions in particular. Deep set functions enforce permutation invariance by using symmetric aggregators [86, 65, 64], which have several applications, *e.g.*, character counting [51], set anomaly detection [51], graph embedding design [34, 47, 80, 71], *etc*. However, they often suffer from limited expressiveness as shown by Wagstaff et al. [82]. Some work aims to overcome this limitations by sequence encoder followed by learning a permutation invariant network structure [62, 68]. However, none of them learns an underlying submodular model in the context of subset selection.

**Learning functions with shape constraints.** Our double-quadrature strategy for concavity is inspired by a series of recent efforts to fit functions with *shape constraints* to suit various learning tasks. Wehenkel and Louppe [83] proposed universal monotone neural networks (UMNN) was a significant early step in learning univariate monotone functions by numerical integration of a non-negative integrand returned by a 'universal' network — this paved the path for universal monotone function modeling. Gupta et al. [33] extended to multidimensional shape constraints for supervised learning tasks, for situations were features complemented or dominated others, or a learnt function $y = f(\boldsymbol{x})$ should be unimodal. Such constraints could be expressed as linear inequalities, and therefore possible to optimize using projected stochastic gradient descent. Gupta et al. [32] widened the scope further to situations where more general constraints had to be enforced on gradients. In the context of density estimation and variational inference, a popular technique is to transform between simple and complex distributions via invertible and differentiable mappings using *normalizing flows* [48], where coupling functions can be implemented as monotone networks. Our work provides a bridge between shape constraints, universal concavity and differentiable subset selection.

**Deep submodular functions (DSF).** Early work predominantly modeled a trainable submodular function as a mixture of fixed submodular functions [53, 77]. If training instances do not fit their 'basis' of hand-picked submodular functions, limited expressiveness results. In the quest for 'universal' submodular function networks, Bilmes and Bai [7] and Bai et al. [3] undertook a thorough theoretical inquiry into the effect of network structure on expressiveness. Specifically, they modeled submodular functions as an aggregate of concave functions of modular functions, computed in a topological order along a directed acyclic graph (DAG), driven by the fact that a concave function of a monotone submodular function is a submodular function [26, 27, 76]. But DSF provides no practical prescription for picking the concave functions. Each application of DSF will need an extensive search over these design spaces.

**Subset selection.** Subset selection especially under submodular or approximate submodular profit enjoys an efficient greedy maximization routine which admits an approximation guarantee. Consequently, a wide variety of set function optimization tasks focus on representing the underlying objective as an instance of a submodular function. At large, subset selection has a myriad of applications in machine learning, e.g., data summarization [4], feature selection [44], influence maximization in social networks [43, 13, 14, 87], opinion dynamics [11, 12, 14, 89, 49], efficient learning [20, 45], human assisted learning [16, 15, 60], etc. However, these works do not aim to learn the underlying submodular function from training subsets.

**Set prediction.** Our work is also related to set prediction. Zhang et al. [90] use a encoder-decoder architecture for set prediction. Rezatofighi et al. [67] provide a deep probabilistic model for set prediction. However, they aim to predict an output set rather than the set function.

**Differentiable subset selection.** Existing trainable subset selection methods [77, 50] often adopt a max-margin optimization approach. However, it requires solving one submodular optimization problem at each epoch, which renders it computationally expensive. On the other hand, Tschiatschek et al. [79] provide a probabilistic soft-greedy model which can generate and be trained on a permutation of subset elements. But then, the trained model becomes sensitive to this specific permutations. Tschiatschek et al. [79] overcome this challenge by presenting several permutations to the learner, which can be inefficient.

**One sided smoothness.** We would like to highlight that our characterization for $\alpha$-submodular function is a special case of one-sided smoothness (OSS) proposed in [29, 28]. However, the significance of these characterizations are different between their and our work. First, they consider $\gamma$-meta submodular function which is a different generalisation of submodular functions compared to $\alpha$-submodular functions. Second, the OSS characterization they provide is for the multilinear extension of $\gamma$-meta submodular function, whereas we provide the characterization of $\alpha$-submodular functions itself, which allows direct construction of our neural models.

**Sample complexity in the context of learning submodular functions.** Goemans et al. [31] provided an algorithm which outputs a function $\hat{f}(S)$ that approximates an arbitrary monotone submodular function $f(S)$ within a factor $O(\sqrt{n}\log n)$ using poly$(n)$ queries on $f$. Their algorithm considers a powerful active probe setting where $f$ can be queried with arbitrary sets. In contrast, Balcan and Harvey [6] consider a more realistic passive setup used in a supervised learning scenario, and designed an algorithm which obtains an approximation of $f(S)$ within factor $O(\sqrt{n})$.

# 3 Design of FLEXSUBNET

In this section, we first present our notations and then propose a family of flexible neural network models for monotone and non-monotone submodular functions and $\alpha$-submodular functions.

## 3.1 Notation and preliminary results

We denote $V = \{1, 2, .., |V|\}$ as the ground set or universal set of elements and $S, T \subseteq V$ as subsets of $V$. Each element $s \in V$ may be associated with a feature vector $\boldsymbol{z}_s$. Given a set function $F : 2^V \to \mathbb{R}$, we define the marginal utility $F(s \,|\, S) := F(S \cup \{s\}) - F(S)$. The function $F$ is called *monotone* if $F(s \,|\, S) \geq 0$ whenever $S \subset V$ and $s \in V \backslash S$; $F$ is called $\alpha$-*submodular* with $\alpha > 0$ if $F(s \,|\, S) \geq \alpha F(s \,|\, T)$ whenever $S \subseteq T$ and $s \in V \backslash T$ [35, 88, 21]. As a special case, $F$ is submodular if $\alpha = 1$ and $F$ is modular if $F(s \,|\, S) = F(s \,|\, T)$. Here, $F(\cdot)$ is called *normalized* if $F(\varnothing) = 0$. Similarly, a real function $f : \mathbb{R} \to \mathbb{R}$ is *normalized* if $f(0) = 0$. Unless otherwise stated, we only consider normalized set functions in this paper. We quote a key result often used for neural modeling for submodular functions [27, 76, 52].

**Proposition 1.** *Given the set function $F : 2^V \to \mathbb{R}^+$ and a real valued function $\phi : \mathbb{R} \to \mathbb{R}$, (i) the set function $\phi(F(\cdot))$ is monotone submodular if $F$ is monotone submodular and $\phi$ is an increasing concave function; and, (ii) $\phi(F(\cdot))$ is non-monotone submodular if $F$ is positive modular and $\phi$ is non-monotone.*

## 3.2 Monotone submodular functions

**Overview.** Our model for monotone submodular functions consists of a neural network which cascades the underlying functions in a recursive manner for $N$ steps. Specifically, to compute the submodular function $F^{(n)}(\cdot)$ at step $n$, it first linearly combines the submodular function $F^{(n-1)}(\cdot)$ computed in the previous step and a *trainable* positive modular function $m^{(n)}(\cdot)$ and then, applies a monotone concave activation function $\phi$ on it.

**Recursive model.** We model the submodular function $F_\theta(\cdot)$ as follows:

$$F^{(0)}(S) = m_\theta^{(0)}(S); \ F^{(n)}(S) = \phi_\theta\big(\lambda F^{(n-1)}(S) + (1-\lambda)m_\theta^{(n)}(S)\big); \ F_\theta(S) = F^{(N)}(S); \quad (1)$$

where the iterations are indexed by $1 \leq n \leq N$, $\{m_\theta^{(n)}(\cdot)\}$ is a sequence of positive modular functions, driven by a neural network with parameter $\theta$. $\lambda \in [0, 1]$ is a tunable or trained parameter. We apply a linear layer with positive weights on the each feature vector $\boldsymbol{z}_s$ to compute the value of $m_\theta^{(n)}(\cdot)$ and then compute $m_\theta^{(n)}(S) = \sum_{s \in S} m_\theta^{(n)}(s)$. Moreover, $\phi_\theta$ is an increasing concave function which, as we shall see later, is modeled using neural networks. Under these conditions, one can use Proposition 1(i) to easily show that $F_\theta(S)$ is a monotone submodular function (Appendix B).

## 3.3 Monotone-$\alpha$-submodular functions

Our characterization for submodular functions in Eq. (1) are based on Proposition 1(i), which implies that a concave composed submodular function is submodular. However, to the best of our knowledge, a similar characterization of $\alpha$-submodular functions is lacking in the literature. To address this gap, we first introduce a novel characterization of monotone $\alpha$-submodular functions and then use it to design a recursive model for such functions.

**Novel characterization of $\alpha$-submodular function.** In the following, we show how we can characterize an $\alpha$-submodular function using a differential inequality (proven in Appendix B).

**Theorem 2.** *Given the function $\varphi : \mathbb{R} \to \mathbb{R}^+$ and a modular function $m : V \to [0, 1]$, the set function $F(S) = \varphi(\sum_{s \in S} m(s))$ is monotone $\alpha$-submodular for $|S| \leq k$, if $\varphi(x)$ is increasing in $x$ and $\frac{\mathrm{d}^2 \varphi(x)}{\mathrm{d}x^2} \leq \frac{1}{k} \log\left(\frac{1}{\alpha}\right) \frac{\mathrm{d}\varphi(x)}{\mathrm{d}x}$.*

The above theorem also implies that given $\alpha = 1$, then $F(S) = \varphi(\sum_{s \in S} m(s))$ is monotone submodular if $\varphi(x)$ is concave in $x$, which reduces to a particular case of Proposition 1(i). Once we design $F(S)$ by applying $\varphi$ on a modular function, our next goal is to design more expressive and flexible modeling in a recursive manner similar to Eq. (1). To this end, we extend Proposition 1 to the case for $\alpha$-submodular functions.

**Proposition 3.** *Given a monotone $\alpha$-submodular function $F(\cdot)$, $\phi(F(S))$ is monotone $\alpha$-submodular, if $\phi(\cdot)$ is an increasing concave function. Here, $\alpha$ remains the same for $F$ and $\phi(F(\cdot))$.*

Note that, when $F(S)$ is submodular, *i.e.*, $\alpha = 1$, the above result reduces to Proposition 1 in the context of concave composed modular functions.

**Recursive model.** Similar to Eq. (1) for submodular functions, our model for $\alpha$-submodular functions is also driven by an recursive model which maintains an $\alpha$-submodular function and updates its value recursively for $N$ iterations. However, in contrast to FLEXSUBNET, where the underlying submodular function is initialized with a positive modular function, we initialize the corresponding $\alpha$-submodular function $F^{(0)}(S)$ with $\varphi_\theta(m_\theta^{(0)}(S))$, where $\varphi_\theta$ is a trainable function satisfying the conditions of Theorem 2. Then we recursively apply a trainable monotone concave function on $F^{(n-1)}(S)$ for $n \in [N]$ to build a flexible model for $\alpha$-submodular function. Formally, we have:

$$F^{(0)}(S) = \varphi_\theta(m_\theta^{(0)}(S)); \; F^{(n)}(S) = \phi_\theta\left(\lambda F^{(n-1)}(S) + (1-\lambda)m_\theta^{(n)}(S)\right); \; F_\theta(S) = F^{(N)}(S);$$
(2)

with $\lambda \in [0, 1]$. Here, $m_\theta$, $\varphi_\theta$ and $\phi_\theta$ are realized using neural networks parameterized by $\theta$. Then, using Proposition 3 and Theorem 2, we can show that $F_\theta(S)$ is $\alpha$-submodular in $S$ (proven in Proposition 6 in Appendix B).

### 3.4 Non-monotone submodular functions

In contrast to monotone set functions, non monotone submodular functions can be built by applying concave function on top of *only one modular function* rather than submodular function (Proposition 1 (i) vs. (ii)). To this end, we model a non-monotone submodular function $F_\theta(\cdot)$ as follows:

$$F_\theta(S) = \psi_\theta(m_\theta(S))$$
(3)

where $m_\theta(\cdot)$ is positive modular function but $\psi_\theta(\cdot)$ can be a *non-monotone* concave function. Both $m_\theta$ and $\psi_\theta$ are trainable functions realized using neural networks parameterized by $\theta$. One can use Proposition 1(ii) to show that $F_\theta(S)$ is a non-monotone submodular function.

### 3.5 Neural parameterization of $m_\theta, \phi_\theta, \psi_\theta, \varphi_\theta$

We complete the neural parameterization introduced in Sections 3.2–3.4. Each model has two types of component functions: (i) the modular set function $m_\theta$; and, (ii) the concave functions $\phi_\theta$ (Eq. (1) and (2)) and $\psi_\theta$ (Eq. (3)) and the non-concave function $\varphi_\theta$ (Eq. (2)). While modeling $m_\theta$ is simple and straightforward, designing neural models for the other components, *i.e.*, $\phi_\theta$, $\psi_\theta$ and $\varphi_\theta$ is non-trivial. As mentioned before, because we cannot rely on the structural complexity of our 'network' (which is a simple linear recursion) or a judiciously pre-selected library of concave functions [7], we need to invest more capacity in the concave function, effectively making it universal.

**Monotone submodular function.** Our model for monotone submodular function described in Eq. (1) consists of two neural components: the sequence of modular functions $\{m_\theta^{(n)}\}$ and $\phi_\theta$.

— *Parameterization of $m_\theta$*. We model the modular function $m_\theta^{(n)} : 2^V \to \mathbb{R}^+$ in Eq. (1) as $m_\theta^{(n)}(S) = \sum_{s \in S} \theta \cdot \boldsymbol{z}_s$, where $\boldsymbol{z}_s$ is the feature vector for each element $s \in S$ and both $\theta, \boldsymbol{z}_s$ are non-negative.

— *Parameterization of $\phi_\theta$*. Recall that $\phi_\theta$ is an increasing concave function. We model it using the fact that a differentiable function is concave if its second derivative is negative. We focus on capturing the second derivative of the underlying function using a complex neural network of arbitrary capacity, providing a negative output. Hence, we use a positive neural network $h_\theta$ to model the second derivative

$$\frac{\mathrm{d}^2\phi_\theta(x)}{\mathrm{d}x^2} = -h_\theta(x) \leq 0.$$
(4)

Now, since $\phi_\theta$ is increasing, we have:

$$\frac{\mathrm{d}\phi_\theta(x)}{\mathrm{d}x} = \int_{b=x}^{b=\infty} h_\theta(b)\,\mathrm{d}b \geq 0 \implies \phi_\theta(x) = \int_{a=0}^{a=x}\int_{b=a}^{b=\infty} h_\theta(b)\,\mathrm{d}b\,\mathrm{d}a.$$
(5)

Here, $\phi_\theta(\cdot)$ is normalized, *i.e.*, $\phi_\theta(0) = 0$, which ensures that $\{F_\theta^{(n)}\}$ in Eq. (1) are also normalized. An offset in Eq. (5) allows a nonzero initial value of $\phi_\theta(\cdot)$, if required. Note that monotonicity and

concavity of $\phi_\theta$ can be achieved by the restricting positivity of $h_\theta$. Such a constraint can be ensured by setting $h_\theta(\cdot) = \text{ReLU}(\Lambda_\theta^{(h)}(\cdot))$, where $\Lambda_\theta^{(h)}(\cdot)$ is any complex neural network. Hence, $\phi_\theta(\cdot)$ represents a class of universal approximators of normalized increasing concave functions, if $\Lambda_\theta^{(h)}(\cdot)$ is an universal approximator of continuous functions [36]. For a monotone submodular function of the form of concave composed modular function, we have the following result (Proven in Appendix B).

**Proposition 4.** *Given an universal set $V$, a constant $\epsilon > 0$ and a submodular function $F(S) = \phi(\sum_{s \in S} m(\boldsymbol{z}_s))$ where $\boldsymbol{z}_s \in \mathbb{R}^d$, $S \subset V$, $0 \leq m(\boldsymbol{z}) < \infty$ for all $\boldsymbol{z} \in \mathbb{R}^d$. Then there exists two fully connected neural networks $m_{\theta_1}$ and $h_{\theta_2}$ of width $d + 4$ and $5$ respectively, each with ReLU activation function, such that the following conditions hold:*

$$\left\| F(S) - \int_{a=0}^{a=\sum_{s \in S} m_{\theta_1}(\boldsymbol{z}_s)} \int_{b=a}^{b=\infty} h_{\theta_2}(b)\, \mathrm{d}b\, \mathrm{d}a. \right\| \leq \epsilon \quad \forall S \subset V \tag{6}$$

**Monotone $\alpha$-submodular model.** An $\alpha$-submodular model described in Eq. (2) has three trainable components: (i) the sequence of modular functions $\{m_\theta^{(n)}(\cdot)\}$, (ii) the concave function $\phi_\theta(\cdot)$ and (iii) $\varphi_\theta(\cdot)$. For the first two components, we reuse the parameterizations used for monotone submodular functions. In the following, we describe our proposed neural parameterization of $\varphi_\theta$.

— *Parameterization of $\varphi_\theta(\cdot)$.* From Theorem 2, we note that $\varphi_\theta(\cdot)$ is increasing and satisfies $\frac{\mathrm{d}^2 \varphi_\theta(x)}{\mathrm{d}x^2} \leq \kappa(\alpha)\frac{\mathrm{d}\varphi_\theta(x)}{\mathrm{d}x}$ where, $\kappa(\alpha) = \frac{1}{k}\log(1/\alpha)$. It implies that

$$e^{-x\kappa(\alpha)}\frac{\mathrm{d}^2\varphi_\theta(x)}{\mathrm{d}x^2} - \kappa(\alpha)e^{-x\kappa(\alpha)}\frac{\mathrm{d}\varphi_\theta(x)}{\mathrm{d}x} \leq 0 \implies \frac{\mathrm{d}}{\mathrm{d}x}\left(e^{-x\kappa(\alpha)}\frac{\mathrm{d}\varphi_\theta(x)}{\mathrm{d}x}\right) \leq 0 \tag{7}$$

Driven by the last inequality, we have

$$e^{-x\kappa(\alpha)}\frac{\mathrm{d}\varphi_\theta(x)}{\mathrm{d}x} = \int_x^\infty g_\theta(b)\, \mathrm{d}b \implies \varphi_\theta(x) = \int_{a=0}^{a=x} e^{a\kappa(\alpha)} \int_{b=a}^{b=\infty} g_\theta(b)\, \mathrm{d}b\, \mathrm{d}a \tag{8}$$

**Parameterizing non-monotone submodular model.** As suggested by Eq. (3), our model for non-monotone submodular function contains a non-monotone concave function $\psi_\theta$ and a modular function $m_\theta$. We model $m_\theta$ using the same parameterization used for the monotone set functions. We parameterize the $\psi_\theta$ as follows.

— *Parameterization of $\psi_\theta$.* Modeling a generic (possibly non-monotone) submodular function requires a general form of concave function $\psi_\theta$ which is not necessarily increasing. The trick is to design $\psi_\theta(\cdot)$ in such a way that its second derivative is negative everywhere, whereas its first derivative can have any sign. For $x \in [0, x_{\max}]$, we have:

$$\psi_\theta(x) = \int_{a=0}^{a=x} \int_{b=a}^{b=\infty} h_\theta(b)\, \mathrm{d}b\, \mathrm{d}a - \int_{a=x_{\max}-x}^{a=x_{\max}} \int_{b=a}^{b=\infty} g_\theta(b)\, \mathrm{d}b\, \mathrm{d}a, \tag{9}$$

where $h_\theta, g_\theta \geq 0$. Moreover, we assume that $\int_0^\infty h_\theta(b)\, \mathrm{d}b$ and $\int_0^\infty g_\theta(b)\, \mathrm{d}b$ are convergent. We use $x_{\max}$ in the upper limit of the second integral to ensure that $\psi_\theta$ is normalized, *i.e.*, $\psi_\theta(0) = 0$. Next, we compute the first derivative of $\psi_\theta(x)$ as:

$$\frac{\mathrm{d}\psi_\theta(x)}{\mathrm{d}x} = \int_{b=x}^{b=\infty} h_\theta(b)\, \mathrm{d}b - \int_{b=x_{\max}-x}^{b=\infty} g_\theta(b)\, \mathrm{d}b, \tag{10}$$

which can have any sign, since both integrals are positive. Here, the second derivative of $\psi_\theta$ becomes

$$\frac{\mathrm{d}^2\psi_\theta(x)}{\mathrm{d}x^2} = -h_\theta(x) - g_\theta(x_{\max} - x) \leq 0 \tag{11}$$

which implies that $\psi_\theta$ is concave. Similar to $h_\theta(\cdot)$, we can model $g_\theta(\cdot) = \text{ReLU}(\Lambda_\theta^g(\cdot))$. Such a representation makes $\psi_\theta(\cdot)$ a universal approximator of normalized concave functions. As suggested by Eq. (3), $\psi_\theta$ takes $m_\theta(S)$ as input. Therefore, in practice, we set $x_{\max} = \max_S m_\theta(S)$.

### 3.6  Parameter estimation from (set, value) pairs

In this section, our goal is to learn $\theta$ from a set of pairs $\{(S_i, y_i) \mid i \in [I]\}$, such that $y_i \approx F_\theta(S_i)$. Hence, our task is to solve the following optimization problem:

$$\min_\theta \sum_{i \in [I]}(y_i - F_\theta(S_i))^2 \tag{12}$$

In the following, we discuss methods to solve this problem.

**Backpropagation through double integral.** Each of our proposed set function models consists of one or more double integrals. Thus computation of the gradients of the loss function in Eq. (12) requires gradients of these integrals. To compute them, we leverage the methods proposed by Wehenkel and Louppe [83], which specifically provide forward and backward procedures for neural networks involving integration. Specifically, we use the Clenshaw-Curtis (CC) quadrature or Trapezoidal Rule for numerical computation of $\phi_\theta(\cdot)$ and $\psi_\theta(\cdot)$. On the other hand, we compute the gradients $\nabla_\theta \phi_\theta(\cdot)$ and $\nabla_\theta \psi_\theta(\cdot)$ by leveraging Leibniz integral rule [24]. In practice, we replace the upper limit ($b = \infty$) of the inner integral in Eqs. (5), (8) and (9) with $b_{max} = x_{max}$ during training.

**Decoupling into independent integrals.** The aforementioned end-to-end training can be challenging in practice, since the gradients are also integrals which can make loss convergence elusive. Moreover, it is also inefficient, since for each step of CC quadrature of the outer integral, we need to perform numerical integration of the entire inner integral. To tackle this limitation, we decouple the underlying double integral into two single integrals, parameterized using two neural networks.

— *Monotone submodular functions.* During training our monotone submodular function model in Eq. (1), we model $\phi_\theta$ in Eq. (5) as:

$$\phi_\theta(x) = \int_0^x \phi_\theta'(a)\, \mathrm{d}a, \quad \phi_\theta'(x) = \int_x^\infty h_\beta(a)\, \mathrm{d}a \tag{13}$$

In contrast to end-to-end training of $\theta$ where $\phi_\theta$ was realized only via neural network of $h_\theta$, here we use two decoupled networks, *i.e.*, $\phi_\theta'$ and $h_\beta$. Then, we learn $\theta$ and $\beta$ by minimizing the regularized sum of squared error, *i.e.*,

$$\min_{\theta,\beta} \sum_{i\in[I]} \sum_{n\in[N]} \left[ \rho\Big(\phi_\theta'(G^{(n)}(S_i)) - \int_{G^{(n)}(S_i)}^\infty h_\beta(a)\, \mathrm{d}a\Big)^2 + \big(y_i - F_\theta(S_i)\big)^2 \right]. \tag{14}$$

Here, $G^{(n)}(S) = \lambda F^{(n-1)}(S) + (1-\lambda)m_\theta^{(n)}(S)$ is the input to $\phi_\theta$ in the recursion (1). Recall that $N$ is the number of steps in the recursion. Since $\phi_\theta$ is monotone, we need $\phi_\theta' \geq 0$ which is ensured by $\phi_\theta'(x) = \mathrm{ReLU}(\Lambda_\theta^{\phi'}(x))$. In principle, we would like to have $\phi_\theta'(x) = \int_x^\infty h_\beta(a)\, \mathrm{d}a$ for all $x \in \mathbb{R}$. In practice, we approximate this by penalizing the regularizer values in the domain of interest—the values which are fed as input to $\phi_\theta$.

While there are potential hazards to such an approximation, in our experiments, the benefits outweighed the risks. The above parameterization involves only single integrals. The use of an auxiliary network $h_\beta$ allows more flexibility, leading to improved robustness during training optimization. The above minimization task achieves approximate concavity of $\phi$ via training, whereas backpropagating through double integrals enforces concavity by design. Appendix C extends this method for $\alpha$-submodular and non-monotone submodular functions.

## 4 Differentiable subset selection

In Section 3.6, we tackled the problem of learning $F_\theta$ from (set, value) pairs. In this section, we consider learning $F_\theta$ from a given set of (perimeter-set, high-value-subset) pair instances.

### 4.1 Learning $F_\theta$ from (perimeter-set, high-value-subset)

Let us assume that the training set $\mathcal{U}$ consists of $\{(V, S)\}$ pairs, where $V$ is some arbitrary subset of the universe of element, and $S \subseteq V$ is a high-value subset of $V$. Given a set function model $F_\theta$, our goal is to estimate $\theta$ so that, across all possible cardinality constrained subsets in $S' \subseteq V$ with $|S'| = |S|$, $F_\theta(\cdot)$ attains its maximum value at $S$. Formally, we wish to estimate $\theta$ so that, for all possible $(V, S) \in \mathcal{U}$, we have:

$$S = \mathrm{argmax}_{S' \subset V} F_\theta(S'), \text{ subject to } |S'| = |S|. \tag{15}$$

### 4.2 Probabilistic greedy model

The problems of maximizing both monotone submodular and monotone $\alpha$-submodular functions rely on a greedy heuristic [59]. It sequentially chooses elements maximizing the marginal gain and hence, cannot directly support backpropagation. Tschiatschek et al. [79] tackle this challenge with a probabilistic model which greedily samples elements from a softmax distribution with the marginal gains as input. Having chosen the first $j$ elements of high-value-subset from perimeter-set $V$, it

draws the $(j+1)^{\text{th}}$ element from the remaining candidates in $V$ with probability proportional to the marginal utility of the candidate. If the elements of $S$ under permutation $\pi$ are written as $\pi(S)_j$ for $j \in [|S|]$, then the probability of selecting the elements of the set $S$ in the sequence $\pi(S)$ is given by

$$\Pr{}_\theta(\pi(S)\,|\,V) = \prod_{j=0}^{|S|-1} \frac{\exp(\tau F_\theta(\pi(S)_{j+1}\,|\,\pi(S)_{\leq j}))}{\sum_{s \in V \setminus \pi(S)_{\leq j}} \exp(\tau F_\theta(s\,|\,\pi(S)_{\leq j}))} \tag{16}$$

Here, $F_\theta$ is the underlying submodular function, $\tau$ is a temperature parameter and $S_0 = \emptyset$.

**Bottleneck in estimating $\theta$.** Note that, the above model (16) generates the elements in a sequential manner and is sensitive to $\pi$. Tschiatschek et al. [79] attempt to remove this sensitivity by maximizing the sum of probability (16) over all possible $\pi$:

$$\theta^* = \underset{\theta}{\mathrm{argmax}} \sum_{(V,S)\in\mathcal{U}} \log \sum_{\pi \in \Pi_{|S|}} \Pr{}_\theta(\pi(S)\,|\,V). \tag{17}$$

However, enumerating all such permutations for even a medium-sized subset is expensive both in terms of time and space. They attempt to tackle this problem by approximating the mode and the mean of $\Pr(\cdot)$ over the permutation space $\Pi_{|S|}$. But, it still require searching over the entire permutation space, which is extremely time consuming.

### 4.3 Proposed approach

Here, we describe our proposed method of permutation adversarial parameter estimation that avoids enumerating all possible permutations of the subset elements, while ensuring that the learned parameter $\theta^*$ remains permutation invariant.

**Max-min optimization problem.** We first set up a max-min game between a permutation generator and the maximum likelihood estimator (MLE) of $\theta$, similar to other applications [68, 62]. Here, for each subset $S$, the permutation generator produces an adversarial permutation $\pi \in \Pi_{|S|}$ which induces a low value of the underlying likelihood. On the other hand, MLE learns $\theta$ in the face of these adversarial permutations. Hence, we have the following optimization problem:

$$\max_\theta \min_{\pi \in \Pi_{|S|}} \sum_{(V,S)\in\mathcal{U}} \log \Pr{}_\theta(\pi(S)\,|\,V) \tag{18}$$

**Differentiable surrogate for permutations.** Solving the inner minimization in (18) requires searching for $\pi$ over $\Pi_{|S|}$, which appears to confer no benefit beyond (17). To sidestep this limitation, we relax the inner optimization problem by using a doubly stochastic matrix $\boldsymbol{P} \in \mathcal{P}_{|S|}$ as an approximation for the corresponding permutation $\pi$. Suppose $\boldsymbol{Z}_S = [\boldsymbol{z}_s]_{s\in S}$ is the feature matrix where each row corresponds to an element of $S$. Then $\boldsymbol{Z}_{\pi(S)} \approx \boldsymbol{P}\boldsymbol{Z}_S$. Thus, $\Pr_\theta(\pi(S)|V)$ can be evaluated by iterating down the rows of $\boldsymbol{P}\boldsymbol{Z}_S$ and, therefore, written in the form $\Pr_\theta(\boldsymbol{P}, S)$. Thus, we get a continuous approximation to (18):

$$\max_\theta \min_{\boldsymbol{P} \in \mathcal{P}_{|S|}} \sum_{(V,S)\in\mathcal{U}} \log \Pr{}_\theta(\boldsymbol{P}, S\,|\,V) \tag{19}$$

so that we can learn $\theta$ by continuous optimization. We generate the soft permutation matrices $\boldsymbol{P}$ using a Gumbel-Sinkhorn network [57]. Given a subset $S$, it takes a seed matrix $\boldsymbol{B}_S$ as input and generates $\boldsymbol{P}^S$ in a recursive manner:

$$\boldsymbol{P}^0 = \exp(\boldsymbol{B}_S/t); \quad \boldsymbol{P}^k = \mathbb{D}_c\left(\mathbb{D}_r\left(\boldsymbol{P}^{(k-1)}\right)\right) \tag{20}$$

Here, $t$ is a temperature parameter; and, $\mathbb{D}_c$ and $\mathbb{D}_r$ provide column-wise and row-wise normalization. Thanks to these two operations, $\boldsymbol{P}^k$ tends to a doubly stochastic matrix for sufficiently large $k$. Denoting $\boldsymbol{P}^\infty = \lim_{k\to\infty} \boldsymbol{P}^k$, one can show [57] that

$$\boldsymbol{P}^\infty = \underset{\boldsymbol{P}\in\mathcal{P}_{|S|}}{\mathrm{argmax}} \mathrm{Tr}(\boldsymbol{P}^\top \boldsymbol{B}_S) - t\sum_{i,j} \boldsymbol{P}(i,j)\log\boldsymbol{P}(i,j)$$

where the temperature $t$ controls the 'hardness' of the "soft permutation" $\boldsymbol{P}$. As $t \to 0$, $\boldsymbol{P}$ tends to be a hard permutation matrix [9]. The above procedure demands different seeds $\boldsymbol{B}_S$ across different pairs of $(V,S) \in \mathcal{U}$, which makes the training computationally expensive in terms of both time and space. Therefore, we model $\boldsymbol{B}_S$ by feeding the feature matrix $\boldsymbol{Z}_S$ into an additional neural network $G_\omega$ which is shared across different $(V,S)$ pairs. Then the optimization (19) reduces to $\max_\theta \min_\omega \sum_{(V,S)\in\mathcal{U}} \log \Pr_\theta(\omega, S)$, with $\omega$ taking the place of $\boldsymbol{P}$.

| | Log | LogDet | FL | Gcut$_{\sigma=0.1}$ | Log×Sqrt | Log×LogDet | Gcut$_{\sigma=0.8}$ |
|---|---|---|---|---|---|---|---|
| FLEXSUBNET | **0.015 ± 0.000** | **0.013 ± 0.000** | **0.022 ± 0.000** | **0.004 ± 0.000** | **0.032 ± 0.000** | **0.025 ± 0.000** | **0.068 ± 0.001** |
| Set-transformer | 0.060 ± 0.001 | 0.029 ± 0.000 | 0.063 ± 0.001 | 0.014 ± 0.000 | 0.037 ± 0.000 | 0.051 ± 0.001 | 0.171 ± 0.002 |
| Deep-set | 0.113 ± 0.002 | 0.044 ± 0.000 | 0.179 ± 0.003 | 0.058 ± 0.001 | 0.079 ± 0.001 | 0.070 ± 0.001 | 0.075 ± 0.001 |
| DSF | 0.258 ± 0.003 | 0.684 ± 0.007 | 0.189 ± 0.003 | 0.778 ± 0.009 | 0.240 ± 0.003 | 0.274 ± 0.003 | 0.970 ± 0.010 |
| SubMix | 0.148 ± 0.002 | 0.063 ± 0.001 | 0.172 ± 0.002 | 0.018 ± 0.000 | 0.158 ± 0.002 | 0.154 ± 0.002 | 1.722 ± 0.015 |

Table 1: Performance measured in terms of RMSE on synthetically generated examples using several set functions. Number in **bold** font (underline) indicate the best (second best) performers.

# 5 Experiments

In this section, we first evaluate FLEXSUBNET using a variety of synthesized set functions and show that it is able to fit them under the supervision of $(set, value)$ pairs more accurately than the state-of-the-art methods. Next, we use datasets gathered from Amazon baby registry [30] to show that FLEXSUBNET can learn to select data from (perimeter-set, high-value-subset) pairs more effectively than several baselines. Our code is in https://tinyurl.com/flexsubnet.

## 5.1 Training by (set, value) pairs

**Dataset generation.** We generate $|V|=10^4$ samples, where we draw the feature vector $z_s$ for each sample $s \in V$ uniformly at random, *i.e.*, $z_s \in \text{Unif}[0, 1]^d$ with $d=10$. Then, we generate subsets $S$ of different sizes by randomly gathering elements from the set $V$. Finally, we compute the values of $F(S)$ for different submodular functions $F$. Specifically, we consider seven planted set functions: (i) **Log**: $F(S) = \log(\sum_{s \in S} \mathbf{1}^\top z_s)$, (ii) **LogDet**: $F(S) = \log \det(\mathbb{I} + \sum_{s \in S} z_s z_s^\top)$ [73, 8], (iii) **Facility location (FL)**: $F(S) = \sum_{s' \in V} \max_{s \in S} z_s^\top z_{s'} / (||z_s|| ||z_{s'}||)$ [58, 25, 19, 61], (iv) **Monotone graph cut (Gcut$_{\sigma=0.1}$)**: In general, Gcut$_\sigma$ is computed using $F(S) := \sum_{u \in V, v \in S} z_u^\top z_v - \sigma \sum_{u,v \in S} z_u^\top z_v$. It measures the weighted cut across $(S, V \setminus S)$ when the weight of the edge $(u, v)$ is computed as $z_u^\top z_v$ [37, 72, 39, 40, 41]. Here, $\sigma$ trades off between diversity and representation. We set $\sigma = 0.1$. Note that Gcut$_\sigma$ is monotone (non-monotone) submodular when $\sigma < 0.5$ ($\sigma > 0.5$). (v) **Log×Sqrt**: $F(S) = [\log(\sum_{s \in S} \mathbf{1}^\top z_s)] \cdot [\sum_{s \in S} \mathbf{1}^\top z_s]^{1/2}$, (vi) **Log×LogDet**: $F(S) = [\log(\sum_{s \in S} \mathbf{1}^\top z_s)] \cdot [\log \det(\mathbb{I} + \sum_{s \in S} z_s z_s^\top)]$ and (vii) **Non monotone graph cut (Gcut$_{\sigma=0.8}$)**: It is the graph cut function in (iv) with $\sigma = 0.8$. Among above set functions, (i)–(iv) are monotone submodular functions, (v)–(vi) are monotone $\alpha$-submodular functions and (vii) is a non-monotone submodular function. We set the number of steps in the recursions (1), (2) as $N = 2$.

**Evaluation protocol.** We sample $|V|=10000$ (set,value) instances as described above and split them into train, dev and test folds of equal size. We present the train and dev folds to the set function model and measure the performance in terms of RMSE on test fold instances. In the first four datasets, we used our monotone submodular model described in Eq. (1); in case of Log×Sqrt and Log×LogDet, we used our $\alpha$-submodular model described in Eq. (2); and, for Gcut$_{\sigma=0.8}$, we used our non-monotone submodular model in Eq. (3). For $\alpha$-submodular model, we tuned $\alpha$ using cross validation. Appendix D provides hyperparameter tuning details for all methods.

**Baselines.** We compare FLEXSUBNET against four state-of-the-art models, *viz.*, Set-transformer [51], Deep-set [86], deep submodular function (DSF) [7] and mixture submodular function (SubMix) [77]. In principle, set transformer shows high expressivity due to its ability to effectively incorporate the interaction between elements. No other method including FLEXSUBNET is able to incorporate such interaction. Thus, given sufficient network depth and width, Set-transformer should show higher accuracy than any other method. However, Set-transformer consumes significantly higher GPU memory even with a small number of parameters. Therefore, to have a fair comparison with the rest non-interaction methods, we kept the parameters of Set-transformer to be low enough so that it consumes same amount of GPU memory, as with other non-interaction based methods.

**Results.** We compare the performance of FLEXSUBNET against the above four baselines in terms of RMSE on the test fold. Table 1 summarizes the results. We make the following observations. (1) FLEXSUBNET outperforms the baselines by a substantial margin across all datasets. (2) Even with reduced number of parameters, Set-transformer outperforms all the other baselines. Set-transformer allows explicit interactions between set elements via all-to-all attention layers. As a result, it outperforms all other baselines even without explicitly using the knowledge of submodularity in its network architecture. Note that, like Deep-set, FLEXSUBNET does not directly model any interaction between the elements. However, it explicitly models submodularity or $\alpha$-submodularity in the network architecture, which helps it outperform the baselines. We observe improvement of the performance of Set-transformer, if we allow more number of parameters (and higher GPU memory).

| | Mean Jaccard Coefficient (MJC) | | | | | | Mean NDCG@10 | | | | | |
|---|---|---|---|---|---|---|---|---|---|---|---|---|
| | FLEXSUBNET | DSF | SubMix | FL | DPP | DisMin | FLEXSUBNET | DSF | SubMix | FL | DPP | DisMin |
| Gear | **0.101** | 0.099 | 0.028 | 0.019 | 0.014 | 0.013 | **0.539** | 0.538 | 0.449 | 0.433 | 0.425 | 0.426 |
| Bath | **0.091** | 0.087 | 0.038 | 0.020 | 0.012 | 0.010 | **0.520** | 0.500 | 0.447 | 0.433 | 0.427 | 0.422 |
| Health | **0.153** | 0.142 | 0.022 | 0.084 | 0.011 | 0.015 | **0.597** | 0.549 | 0.449 | 0.540 | 0.425 | 0.435 |
| Diaper | **0.134** | 0.115 | 0.023 | 0.018 | 0.013 | 0.012 | **0.562** | 0.546 | 0.447 | 0.440 | 0.435 | 0.435 |
| Toys | **0.157** | 0.150 | 0.025 | 0.064 | 0.029 | 0.029 | **0.591** | 0.577 | 0.446 | 0.472 | 0.448 | 0.449 |
| Bedding | **0.203** | 0.191 | 0.028 | 0.015 | 0.043 | 0.047 | **0.643** | 0.623 | 0.437 | 0.438 | 0.456 | 0.461 |
| Feeding | **0.100** | 0.091 | 0.026 | 0.023 | 0.020 | 0.019 | **0.550** | 0.547 | 0.459 | 0.453 | 0.454 | 0.452 |
| Apparel | **0.101** | 0.093 | 0.036 | 0.022 | 0.016 | 0.016 | **0.558** | 0.550 | 0.459 | 0.452 | 0.446 | 0.444 |
| Media | **0.135** | 0.130 | 0.029 | 0.035 | 0.029 | 0.025 | **0.578** | 0.578 | 0.474 | 0.470 | 0.461 | 0.461 |

Table 2: Prediction of subsets in product recommendation task. Numbers in **bold** (underline) indicate best (second best) performer.

(3) While, in principle, DSF function family contains that of FLEXSUBNET, standard multilayer instantiations of DSF with fixed concave functions cannot learn well from (set,value) training.

### 5.2 Training by (perimeter-set, high-value-subset)

**Datasets.** We use the Amazon baby registry dataset [30] which contains 17 product categories. Among them, we only consider those categories where $|V| > 50$, where $V$ is the total number of items in the universal set. These categories are: (i) Gear, (ii) Bath, (iii) Health, (iv) Diaper, (v) Toys, (vi) Bedding, (vii) Feeding, (viii) Apparel and (ix) Media. They are also summarized in Appendix F.

**Evaluation protocol.** Each dataset contains a universal set $V$ and a set of subsets $\mathcal{S} = \{S\}$. Each item $s$ is characterized by a short textual description such as "bath: Skip Hop Moby Bathtub Elbow Rest, Blue : Bathtub Side Bumpers : Baby". From this text, we compute $z_s$ using BERT [18]. We split $\mathcal{S}$ into equal-sized training ($\mathcal{S}_{\text{train}}$), dev ($\mathcal{S}_{\text{dev}}$) and test ($\mathcal{S}_{\text{test}}$) folds. We disclose the training and dev sets to the submodular function models, which are trained using our proposed permutation adversarial approach (Section 4). Then the trained model is used to sample item sequence $S'$ with prefix $S'_{\leq K} = \{s_1, \ldots, s_K\}$. Here, $s_i$ is the item selected at the $i^{\text{th}}$ step of the greedy algorithm. We assess the quality of the sampled sequence using two metrics.

*Mean Jaccard coefficient (MJC).* Given a set $T$ in the test set, we first measure the overlap between $T$ and the first $|T|$ elements of $S'$ using Jaccard coefficient, *i.e.*, $JC(T) = |S'_{\leq |T|} \cap T| / |S'_{\leq |T|} \cup T|$ and then average over all subsets in the test set, *i.e.*, $T \in \mathcal{S}_{\text{test}}$ to compute mean Jaccard coefficient [38].

*Mean NDCG@10.* The greedy sampler outputs item sequence $S'$. For each $T \in \mathcal{S}_{\text{test}}$, we compute the NDCG given by the order of first 10 elements of $S'$, where $i \in S'$ is assigned a gold relevance label 1, if $i \in T$ and 0, otherwise. Finally, we average over all test subsets to compute Mean NDCG@10.

**Our method vs. baselines.** We compare the subset selection ability of FLEXSUBNET against several baselines which include two trainable submodular models: (i) Deep submodular function (DSF) and (ii) Mixture of submodular functions (SubMix) described in Section 5.1; two popular non-trainable submodular functions which include (iii) Facility location (FL) [58, 25], (iv) Determinantal point process (DPP) [8]; and, a non-submodular function (v) Disparity Min (DisMin) which is often used in data summarization [10]. Here, we use the monotone submodular model of FLEXSUBNET. Appendix F contains more details about the baselines. We did not consider general purpose set functions, *e.g.*, Set-transformer, Deep-set, etc., because they cannot be maximized using greedy-like algorithms and therefore, we cannot apply our proposed method in Section 4.

**Results.** Table 2 provides a comparative analysis across all candidate set functions, which shows that: (i) FLEXSUBNET outperforms all the baselines across all the datasets; (ii) DSF is the second best performer across all datasets; and, (iii) the performance of non-trainable set functions is poor, as they are *not trained to mimic* the set selection process.

## 6 Conclusion

We introduced FLEXSUBNET: a family of submodular functions, represented by neural networks that implement quadrature-based numerical integration, and supports end-to-end backpropagation through these integration operators. We designed a permutation adversarial subset selection method, which ensures that the estimated parameters are independent of greedy item selection order. On both synthetic and real datasets, FLEXSUBNET improves upon recent competitive formulations. Our work opens up several avenues of future work. One can extend our work for $\gamma$-weakly submodular functions [22]. Another extension of our work is to leverage other connections to convexity, *e.g.*, the Lovasz extension [54, 1] similar to Karalias et al. [42].

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
