# Neural Estimation of Submodular Functions with Applications to Differentiable Subset Selection (Appendix)

## A   Potential limitations of our work

One of the key limitations of our work is that our neural models are limited to modeling concave composed modular functions. However the class of submodular functions are larger. One of the way to address this problem to model a submodular function using Lovasz extension [54, 85]. Another key limitation is our approach cannot model weakly submodular functions at large, which is superset of approximate submodular functions modeled here. We would like to extend our work in this context. Our differentiable subset selection method does not have access to supervision of set values. Thus training only from high value subset can lead to high bias towards a specific set of elements. An interesting direction would be to mitigate such bias.

## B   Proofs of the technical results for Section 3

### B.1   Formal justification of the submodularity of FLEXSUBNET given by Eq. (1)

**Proposition 5.** *Let $m_\theta^{(n)} : 2^V \to \mathbb{R}^+$ be a modular function, i.e., $m_\theta^{(n)}(S) = \sum_{s \in S} m_\theta^{(n)}(\{s\})$; $\phi_\theta$ be a monotone concave function. Then, $F_\theta(S)$ computed using Eq. (1) is monotone submodular.*

*Proof.* We proof this by induction. Clearly, $F^{(0)}$ is monotone submodular. Now, assume that $F^{(n-1)}(S)$ is monotone submodular. Then, $R(S) = \lambda \cdot F_\theta^{(n-1)}(S) + (1 - \lambda) \cdot m_\theta^{(n)}(S)$ is monotone submodular. Hence, from Proposition 1 (i), we have $F_\theta^{(n)}(S) = \phi_\theta(R(S))$ to be submodular.   □

### B.2   Proof of Theorem 2

**Theorem 2.** *Given the functions $\varphi : \mathbb{R} \to \mathbb{R}^+$ and $m : V \to [0, 1]$. Then, the set function $F(S) = \varphi(\sum_{s \in S} m(\{s\}))$ is monotone $\alpha$-submodular for $|S| \le k$, if $\varphi(x)$ is increasing in $x$ and*

$$\frac{\partial^2 \varphi(x)}{\partial x^2} \le \frac{1}{k} \log\left(\frac{1}{\alpha}\right) \frac{\partial \varphi(x)}{\partial x} \tag{21}$$

*Proof.* Since, both $\varphi$ and $F$ is monotone, $\varphi \circ F$ is monotone. Shifting $x$ to $x + y$ with $y > 0$, we have:

$$\frac{\partial^2 \varphi(x + y)}{\partial y^2} \le \kappa(\alpha) \frac{\partial \varphi(x + y)}{\partial y} \tag{22}$$

where, $\kappa(\alpha) = \frac{1}{k} \log\left(\frac{1}{\alpha}\right)$. Eq. (22) implies that

$$e^{-y\kappa(\alpha)} \frac{\partial^2 \varphi(x + y)}{\partial y^2} - \kappa(\alpha) e^{-y\kappa(\alpha)} \frac{\partial \varphi(x + y)}{\partial y} \le 0 \implies \frac{\partial}{\partial y}\left(\frac{e^{-y\kappa(\alpha)} \partial \varphi(x + y)}{\partial y}\right) \le 0 \tag{23}$$

Hence, $\dfrac{e^{-y\kappa(\alpha)} \partial \varphi(x + y)}{\partial y}$ is decreasing function in $y$. Hence,

$$\frac{e^{-y\kappa(\alpha)} \partial \varphi(x + y)}{\partial y} \le \left.\frac{e^{-y\kappa(\alpha)} \partial \varphi(x + y)}{\partial y}\right|_{y=0} = \frac{\partial \varphi(x)}{\partial x} \tag{24}$$

Next, we define $\varphi_S(\bullet) = \varphi(\bullet + m(S))$ for all $S$ and then we compute

$$\frac{\varphi(m(S \cup s)) - \varphi(m(S))}{\varphi(m(T \cup s)) - \varphi(m(T))} = \frac{\varphi(m(s) + m(S)) - \varphi(m(S))}{\varphi(m(s) + m(T)) - \varphi(m(T))} \qquad \text{(Since } m \text{ is modular)} \qquad (25)$$

$$= \frac{\varphi_S(m(s)) - \varphi_S(0)}{\varphi_T(m(s)) - \varphi_T(0)} \qquad (26)$$

$$= \frac{\left.\dfrac{\partial \varphi_S(x)}{\partial x}\right|_{x=c}}{\left.\dfrac{\partial \varphi_T(x)}{\partial x}\right|_{x=c}} \qquad \text{(for some } c \in (0, m(s)); \text{ Cauchy's mean value Theorem)}$$

$$\geq \exp(-\kappa(\alpha)[m(T) - m(S)]) \qquad \text{(Using Eq. (24))}$$

$$\geq \exp(-\log(1/\alpha)) \qquad \text{(Since } |S|, |T| \leq k) \qquad (27)$$

$\square$

## B.3 Proof of proposition 3

**Proposition 3.** *Given a monotone $\alpha$-submodular function $F(\bullet)$. Then, $\phi(F(S))$ is monotone $\alpha$-submodular, if $\phi(\bullet)$ is an increasing concave function.*

*Proof.* Assume $S \subset T$ and $s \in V \backslash T$. Since $F(\bullet)$ is monotone, we have $F(S) \leq F(T)$ and $F(S \cup s) \leq F(T \cup s)$. Using concavity of $\varphi$ and comparing the slope of two chords, we have:

$$\frac{\varphi(F(S \cup s)) - \varphi(F(S))}{F(S \cup s) - F(S)} \geq \frac{\varphi(F(T \cup s)) - \varphi(F(T))}{F(T \cup s) - F(T)}$$

$$\implies \varphi(F(S \cup s)) - \varphi(F(S)) \geq \frac{F(S \cup s) - F(S)}{F(T \cup s) - F(T)} [\varphi(F(T \cup s)) - \varphi(F(T))] \qquad (28)$$

$F$ is monotone. Hence, the last inequality is obtained by multiply both sides by the positive quantity $F(S \cup s) - F(S)$ which keeps the sign of the inequality unchanged.

Now, since $\varphi$ is an increasing function and $F$ is monotone, $\varphi(F(T \cup s)) - \varphi(F(T)) \geq 0$. Then, since $F$ is monotone $\alpha$-submodular, we have $\frac{F(S\cup s)-F(S)}{F(T\cup s)-F(T)} \geq \alpha$. Hence, we have $\frac{F(S\cup s)-F(S)}{F(T\cup s)-F(T)} [\varphi(F(T \cup s)) - \varphi(F(T))] \geq \alpha[\varphi(F(T \cup s)) - \varphi(F(T))]$. Together with Eq. (28), it finally gives

$$\varphi(F(S\cup s)) - \varphi(F(S)) \geq \frac{F(S \cup s) - F(S)}{F(T \cup s) - F(T)} [\varphi(F(T \cup s)) - \varphi(F(T))] \geq \alpha[\varphi(F(T\cup s)) - \varphi(F(T))].$$

$\square$

## B.4 Proof of Proposition 4

**Proposition 4.** *Given an universal set $V$, a constant $\epsilon > 0$ and a submodular function $F(S) = \phi(\sum_{s \in S} m(\boldsymbol{z}_s))$ where $\boldsymbol{z}_s \in \mathbb{R}^d$, $S \subset V$, $0 \leq m(\boldsymbol{z}) < \infty$ for all $\boldsymbol{z} \in \mathbb{R}^d$. Note that here, $F$ is normalized and therefore, $F(\emptyset) = \phi(0) = 0$. Then there exists two fully connected neural networks $m_{\theta_1}$ and $h_{\theta_2}$ of width $d + 4$ and $5$ respectively, each with ReLU activation function, such that the following conditions hold:*

$$\left| F(S) - \int_{a=0}^{a=\sum_{s \in S} m_{\theta_1}(\boldsymbol{z}_s)} \int_{b=a}^{b=\infty} h_{\theta_2}(b) \, \mathrm{d}b \, \mathrm{d}a. \right| \leq \epsilon \quad \forall S \subset V \qquad (29)$$

*Proof.* Since our functions are normalized, we have $F(\emptyset) = 0 = \phi(0)$. Assume $d\phi(b)/db \to 0$ as $b \to \infty$. Note that the condition that $\lim_{b \to \infty} d\phi(b)/db \to 0$ is not a restriction since the maximum value of $x$ that goes as input to provide output $\phi(x)$ is $x_{\max} = \sum_{s \in V} m(\boldsymbol{z}_s)$ which is finite. Therefore, one can always define $\phi(\bullet)$ outside that regime ($x > x_{\max}$) as constant zero. Then we can write:

$$\phi(x) = \int_{a=0}^{a=x} \int_{b=a}^{b=\infty} \left[ -\frac{d^2\phi(b)}{db^2} \right] \mathrm{d}b \, \mathrm{d}a \qquad (30)$$

To prove the above, one can define the RHS of Eq. (30) as say, $\phi_1(x)$. We note that $\phi(0) = 0$ because it is normalized. Thus, we have:

$$\phi_1(0) = \phi(0) \tag{31}$$

$$\frac{d\phi_1(x)}{dx} = \frac{d\phi(x)}{dx} \quad \text{(Since, } \lim_{b\to\infty} d\phi(b)/db \to 0) \tag{32}$$

These two conditions give us: $\phi(x) = \phi_1(x)$. Now, we can write $F(S)$ as follows:

$$F(S) = \int_{a=0}^{a=\sum_{s\in S} m(\boldsymbol{z}_s)} \int_{b=a}^{b=\infty} \left[ -\frac{d^2\phi(b)}{db^2} \right] \, db \, da \tag{33}$$

Let us define $h(b) = -\frac{d^2\phi(b)}{db^2}$. Choose $\epsilon_m > 0$ and $\epsilon_h > 0$. According to [55, Theorem 1], we can say that it is possible to find ReLU neural networks $m_{\theta_1}$ and $h_{\theta_2}$ for widths $d + 4$ and $5$ respectively, for which we have,

$$\int_{\mathbb{R}^d} |m(\boldsymbol{z}) - m_{\theta_1}(\boldsymbol{z})| d\boldsymbol{z} < \epsilon_m \tag{34}$$

$$\int_{\mathbb{R}^+} |h(b) - h_{\theta_2}(b)| db < \epsilon_h \tag{35}$$

For continuous functions $m$, the first condition suggest that there exists $\epsilon'_m$ for which $|m(\boldsymbol{z}) - m_{\theta_1}(\boldsymbol{z})| \le \epsilon'_m$.

Then we have the following:

$$F(S) - \int_{a=0}^{a=\sum_{s\in S} m_{\theta_1}(\boldsymbol{z}_s)} \int_{b=a}^{b=\infty} h_{\theta_2}(b) \, db \, da \tag{36}$$

$$= \int_{a=0}^{a=\sum_{s\in S} m(\boldsymbol{z}_s)} \int_{b=a}^{b=\infty} h(b) \, db \, da - \int_{a=0}^{a=\sum_{s\in S} m_{\theta_1}(\boldsymbol{z}_s)} \int_{b=a}^{b=\infty} h(b) \, db \, da \tag{37}$$

$$+ \int_{a=0}^{a=\sum_{s\in S} m_{\theta_1}(\boldsymbol{z}_s)} \int_{b=a}^{b=\infty} h(b) \, db \, da - \int_{a=0}^{a=\sum_{s\in S} m_{\theta_1}(\boldsymbol{z}_s)} \int_{b=a}^{b=\infty} h_{\theta_2}(b) \, db \, da \tag{38}$$

$$= \int_{a=\sum_{s\in S} m_{\theta_1}(\boldsymbol{z}_s)}^{a=\sum_{s\in S} m(\boldsymbol{z}_s)} \int_{b=a}^{b=\infty} h(b) \, db \, da + \int_{a=\sum_{s\in S} m(\boldsymbol{z}_s)}^{a=\sum_{s\in S} m_{\theta_1}(\boldsymbol{z}_s)} \int_{b=a}^{b=\infty} [h(b) - h_{\theta_2}(b)] \, db \, da \tag{39}$$

$$+ \int_{a=\sum_{s\in S} m(\boldsymbol{z}_s)}^{0} \int_{b=a}^{b=\infty} [h(b) - h_{\theta_2}(b)] \, db \, da \tag{40}$$

This gives us:

$$\left| F(S) - \int_{a=0}^{a=\sum_{s\in S} m_{\theta_1}(\boldsymbol{z}_s)} \int_{b=a}^{b=\infty} h_{\theta_2}(b) \, db \, da \right| \tag{41}$$

$$\le |V| \epsilon'_m \int_{b=0}^{b=\infty} h(b) \, db \, da + |V| \epsilon'_m \epsilon_h + +|V| m_{\max} \epsilon_h. \tag{42}$$

Here, $m_{\max} = \max_{s\in V} m(\boldsymbol{z}_s)$. One can choose $\epsilon_\bullet$ in order to set the RHS to $\epsilon$. $\qquad\square$

## B.5 Formal result showing that $F_\theta$ computed using Eq. (2) is $\alpha$-submodular

**Proposition 6.** *Let $m_\theta^{(n)} : 2^V \to \mathbb{R}^+$ be a modular function, i.e., $m_\theta^{(n)}(S) = \sum_{s\in S} m_\theta^{(n)}(\{s\})$; $\varphi_\theta(\bullet)$ satisfy the conditions of Theorem 2 with $\alpha \in (0,1)$; $\phi_\theta$ be a monotone concave function. Then, $F_\theta(S)$ computed using Eq. (2) is a monotone $\alpha$-submodular function.*

*Proof.* We proof this by induction. From Theorem 2, $F^{(0)}$ is monotone $\alpha$-submodular. Now, assume that $F^{(n-1)}(S)$ is monotone $\alpha$-submodular. Then, we see that $R(S) = \lambda \cdot F^{(n-1)}(S) + (1-\lambda) \cdot m_\theta^{(n)}(S)$ is monotone. To prove $R(S)$ is $\alpha$-submodular, we proceed as follows:

$$R(s \mid S) = \lambda \cdot F^{(n-1)}(s \mid S) + (1-\lambda) \cdot m_\theta^{(n)}(s)$$

$$\ge \lambda\alpha \cdot F^{(n-1)}(s \mid T) + \alpha(1-\lambda) \cdot m_\theta^{(n)}(s) \quad (F(S) \text{ is } \alpha\text{-submodular}, \alpha m_\theta^{(n)}(s) < m_\theta^{(n)}(s))$$

$$\ge \alpha R(s \mid T). \tag{43}$$

Applying Proposition 3 on the above leads to the final result.

# C Decoupling integrals for $\alpha$-submodular and non-monotone submodular functions

— *Monotone $\alpha$-submodular function.* In case of monotone $\alpha$-submodular functions, we first break the double integral of $\phi_\theta$ used in the recursion (2) using the two single integrals described in Eq. (13). In addition, we need to model $\varphi_\theta$ in Eq. (8) as follows:

$$\varphi_\theta(x) = \int_0^x \varphi_\theta'(a)\, \mathrm{d}a, \ \ \varphi_\theta'(x) = \int_x^\infty e^{a\kappa(\alpha)} g_\gamma(a)\, \mathrm{d}a \tag{44}$$

Then, we apply two regularizers with the loss (12): one for $\phi_\theta'$ similar to Eq. (14) and the other for the second integral equation in Eq. (44) which is computed as $\sum_{i\in[I]} \rho\left(\varphi_\theta'(m_\theta^{(0)}(S_i)) - \int_{m_\theta^{(0)}(S_i)}^\infty e^{a\kappa(\alpha)} g_\gamma(a)\, \mathrm{d}a\right)^2$. Then, we minimize the regularized loss with respect to $\theta, \beta, \gamma$.

— *Non-monotone submodular function.* In case of non-monotone submodular function models, we decouple $\psi_\theta$ in Eq. (9) into the following set of intergals:

$$\psi_\theta(x) = \int_0^x \psi_{h,\theta}'(a)\mathrm{d}a - \int_{x_{\max}-x}^{x_{\max}} \psi_{g,\theta}'(a)\, \mathrm{d}a \tag{45}$$

$$\psi_{h,\theta}'(x) = \int_x^\infty h_\beta(b),\mathrm{d}b, \quad \psi_{g,\theta}'(x) = \int_x^\infty g_\gamma(b)\, \mathrm{d}b \tag{46}$$

Then, we add the regularizers corresponding to the last two integral equations (46) to the loss (12) and then minimize it to train $\theta, \beta, \phi$.

# D Additional details about experiments on learning from (set, value) pairs

## D.1 Additional details about the data generation

As mentioned in Section 5.1, we generate $|V| = 10^4$ items. We draw the feature vector $\boldsymbol{z}_s$ for each item $s \in V$ uniformly at random *i.e.*, $\boldsymbol{z}_s \in \mathrm{Unif}[0,1]^d$, where $d = 10$. Here, we use such a generative process for the features since our synthetic set functions often require positive input. Then, we generate subsets $S$ of different sizes by gathering elements from the universal set $V$ as follows. We randomly shuffle the elements of $V$ to obtain a sequence of elements $\{s_1, ..., s_{|V|}\}$. We construct $|V|$ subsets by gathering top $j$ elements for $j = 1, .., |V|$, *i.e.*, $\mathcal{S} = \{S\} = \{\{s_1, ..., s_j\} \,|\, j \leq |V|\}$.

## D.2 Implementation details of FLEXSUBNET

We model the integrands $h_\theta$ of the submodular funcions using one input layer, three hidden layers and one output layer, each with width 50. Here, the input and hidden layers are built using one linear and ReLU units and the output layer is an ELU activation unit. For forward and backward passes under integration, we adapt the code provided by Wehenkel and Louppe [83] in our setup. Moreover we choose the modular function $m_\theta^{(n)}(\{s\}) = \theta_m^\top \boldsymbol{z}_s$. We set the value of maximum number of steps using cross validation, which gives $N = 2$ for all datasets. We set the value of weight decay as $1e - 4$ and learning rate as $2e - 3$.

## D.3 Implementation details of the baselines

**Set-transformer [51].** Our implementation of Set-transformer has one input layer, two hidden layers and one output layer. We would like to hightlight that with increased number of parameters, Set-transformer consumed large GPU memory (max GPU usage > 8GB for even 29 parameters). This is because of two reasons: (1) the set transformer performs all-to-all attention architecture and (2) the size of universe in our experiments is $|V| = 10000$. Set-transformer involves several concatenation operation which blows up the intermediate tensors.

**Deep set [86].** Our implementation of deep set model is similar to the integrand of FLEXSUBNET, *i.e.*, it consists of one input layer, three hidden layers and one output layer, each with width 50. Here, the input and hidden layers is built of one linear and ReLU units whereas, the output layer is an ELU activation unit. Here, we set the value of weight decay as $10^{-4}$ and learning rate as $2\times10^{-3}$.

**Deep submodular function (DSF) [7].** DSF makes no prescription about the choice of network depth, width, or concave functions. Other researchers commonly make simple choices such as fully-connected layers with arbitrary concave activation [56]. Similarly, we use the monotone concave function $\phi_\theta(x) = \log(x + \theta)$ with trained $\theta \in \mathbb{R}_+$ and the modular function $m_\theta(s) = \theta_m^\top z_s$. Similar to our method, we use the network depth $N = 2$ for DSF. In our experiments, we found that not using the offset gave unacceptably poor predictive performance. We set the value of weight decay as $10^{-4}$ and learning rate as $10^{-3}$.

**Mixture submodular function (SubMix) [77].** Here, we consider $F_\theta(S) = \theta_1 \log(\sum_{s \in S} \theta_m^\top z_s) + \theta_2 \log\log(\sum_{s \in S} \theta_m^\top z_s) + \theta_3 \log\log\log(\sum_{s \in S} \theta_m^\top z_s)$. Note that this design does not make sure $\log(\sum_{s \in S} \theta_m^\top z_s) > 0$ or $\log\log(\sum_{s \in S} \theta_m^\top z_s) > 0$. However, with valid initial conditions $\theta_{m,0}$ where $\log(\sum_{s \in S} \theta_{m,0}^\top z_s) > 0$ and $\log\log(\sum_{s \in S} \theta_{m,0}^\top z_s) > 0$ and the current learning rate $10^{-3}$, we observed that $\theta_m$ always ensured that $\log(\sum_{s \in S} \theta_m^\top z_s) > 0$ and $\log\log(\sum_{s \in S} \theta_m^\top z_s) > 0$ throughout our training.

Initially we started with $F_\theta(S) = \theta_1 \log(1 + \sum_{s \in S} \theta_m^\top z_s) + \theta_2 \log(1 + \log(1 + \sum_{s \in S} \theta_m^\top z_s)) + \theta_3 \log(1 + (\log(1 + \log(1 + \sum_{s \in S} \theta_m^\top z_s))))$, which always would ensure that $\log(\sum_{s \in S} \theta_m^\top z_s) > 0$ and $\log\log(\sum_{s \in S} \theta_m^\top z_s) > 0$. But we observed that the performance deteriorates than the current candidate which does not add 1 to each log term.

Moreover, we observed that adding additional component did not improve accuracy. Here, we set the value of weight decay as $10^{-4}$ and learning rate as $10^{-3}$.

In all models, we set the initial value of the parameter vector of the modular function to be $\theta_m = \mathbf{1}$ which ensured that the final trained model $\theta_m \geq 0$. In all experiments, we set the batch size as 66. For each model, we train for 400 epochs and choose the training model which shows the best performance in last 10 epochs. We choose the best initial model based on the performance of final trained model on the validation set.

### D.4  Computation of $\alpha$ in synthetically planted functions

We define the curvature $F$ [81] as:

$$\text{curv}_F = 1 - \min_{S, j \notin S} \frac{F(j \mid S)}{F(j \mid \emptyset)} \tag{47}$$

Define $z_{\max} = \max_{s \in V} ||z||_\infty$, $z_{\min} = \min_{s \in V, i \in [d]}$ and assume $z_{\min} > e^{-2}$.

**Log $\times$ LogDet.** First we consider $F(S) = [\log(\sum_{s \in S} \mathbf{1}^\top z_s)] \cdot [\log \det(\mathbb{I} + \sum_{s \in S} z_s z_s^\top)]$. Assume that

$$f(S) = \log(\sum_{s \in S} \mathbf{1}^\top z_s)$$
$$g(S) = \log \det(\mathbb{I} + \sum_{s \in S} z_s z_s^\top) \tag{48}$$

Now, we have:

$$\begin{aligned}
F(S \cup k) - F(S) &= f(S \cup k)\, g(S \cup k) - f(S)\, g(S) \\
&= f(S \cup k)(g(S \cup k) - g(S)) + g(S)(f(S \cup k) - f(S)) \\
&\geq f_{\min}\, \text{curv}_g\, g_{\min}
\end{aligned} \tag{49}$$

Similarly,

$$\begin{aligned}
F(T \cup k) - F(T) &= f(T \cup k)\, g(T \cup k) - f(T)\, g(T) \\
&= f(T \cup k)(g(T \cup k) - g(T)) + g(T)(f(T \cup k) - f(T))
\end{aligned} \tag{50}$$

Now, $\log \det(A) \leq \text{tr}(A - \mathbb{I})$. Hence, we have:

$$g(T \cup k) - g(T) = \log \det \left( \mathbb{I} + \left( \mathbb{I} + \sum_{s \in T} z_s z_s^\top \right)^{-1} z_k z_k^\top \right)$$

$$\leq z_k^\top \left( \mathbb{I} + \sum_{s \in T} z_s z_s^\top \right)^{-1} z_k \tag{51}$$

Moreover, $f(T \cup k) = \log(\sum_{s \in T \cup k} \mathbf{1}^\top \mathbf{z}_s) \leq \sum_{s \in T \cup k} \mathbf{1}^\top \mathbf{z}_s$. Hence, $f(T \cup k)(g(T \cup k) - g(T))$ satisfies:

$$f(T \cup k)(g(T \cup k) - g(T)) \leq \mathbf{z}_k^\top \left( \frac{1}{\sum_{s \in T \cup k} \mathbf{1}^\top \mathbf{z}_s} \left( \mathbb{I} + \sum_{s \in T} \mathbf{z}_s \mathbf{z}_s^\top \right) \right)^{-1} \mathbf{z}_k. \tag{52}$$

Here, we make some crude probabilistic argument. Since $\mathbf{z}_s$ is iid uniform random variables, $\sum_{s \in T} \mathbf{z}_s \mathbf{z}_s^\top \approx |T| \left[ \mathbb{I}/12 + \mathbf{1}\mathbf{1}^T/4 \right]$ and

$$f(T \cup k)(g(T \cup k) - g(T)) \leq 24 d^3 z_{\max}^3 \tag{53}$$

The second term in Eq. (50) shows that:

$$g(T)(f(T \cup k) - f(T)) \leq \log \det \left( \mathbb{I} + \sum_{s \in T} \mathbf{z}_s \mathbf{z}_s^\top \right) \cdot \log \left( 1 + \frac{\mathbf{1}^\top \mathbf{z}_k}{\sum_{s \in T} \mathbf{1}^\top \mathbf{z}_s} \right)$$

$$\leq \operatorname{tr} \left( \sum_{s \in T} \mathbf{z}_s \mathbf{z}_s^\top \right) \cdot \frac{\mathbf{1}^\top \mathbf{z}_k}{\sum_{s \in T} \mathbf{1}^\top \mathbf{z}_s} \leq d^2 z_{\max}^2 / z_{\min}^2 \tag{54}$$

Hence, $F(S)$ is $\alpha$-submodular with

$$\alpha > \alpha^* = \frac{f_{\min} \max\{\operatorname{curv}_g, \operatorname{curv}_f\} g_{\min}}{d^2 z_{\max}^2 / z_{\min}^2 + 24 d^3 z_{\max}^3}$$

**Log $\times$ Sqrt.** Now consider $F(S) = \sum_{s \in S} \log \left( \mathbf{1}^\top \mathbf{z}_s \right) \sqrt{\mathbf{1}^\top \mathbf{z}_s}$. By mean value theorem:

$$F(S \cup k) - F(S) = (\mathbf{1}^\top \mathbf{z}_k) \frac{d}{dx} \log x \sqrt{x} \Big|_{x \in (\sum_{s \in S} \mathbf{1}^\top \mathbf{z}_s, \sum_{s \in S \cup k} \mathbf{1}^\top \mathbf{z}_s)}$$

$$= (\mathbf{1}^\top \mathbf{z}_k) \frac{2 + \log x}{2\sqrt{x}} \tag{55}$$

Similarly $F(T \cup k) - F(T) = (\mathbf{1}^\top \mathbf{z}_k) \max_y \frac{2 + \log y}{2\sqrt{y}}$ where $y \in (\sum_{s \in T} \mathbf{1}^\top \mathbf{z}_s, \sum_{s \in T \cup k} \mathbf{1}^\top \mathbf{z}_s)$.

$$\alpha \geq \frac{\frac{2 + \log x}{2\sqrt{x}}}{\frac{2 + \log y}{2\sqrt{y}}} \geq \frac{2 + \log z_{\min}}{2\sqrt{z_{\min}}} \tag{56}$$

The above is due to the fact that: $\max_y \frac{2 + \log y}{2\sqrt{y}} = 1$ at $y = 1$.

# E  Additional experiments with synthetic data

|  | Log | LogDet | FL |
|---|---|---|---|
| Decoupling | **0.015 ± 0.000** | **0.013 ± 0.000** | **0.022 ± 0.000** |
| End-end | 0.078 ± 0.001 | 0.073 ± 0.001 | 0.078 ± 0.001 |
| Our ($N = 1$) | 0.089 ± 0.001 | 0.075 ± 0.001 | 0.089 ± 0.001 |

Table 3: Variants of our approach.

| FLEXSUBNET | Set-transformer | Deep-set | DSF | SubMix |
|---|---|---|---|---|
| 0.055 | 0.119 | 0.160 | 2.31 | 1.770 |

Table 4: Performance measured in terms of RMSE on synthetically generated examples using $F(S) = \min(\sum_{s \in S} \mathbf{1}^\top \mathbf{z}_s, b + \min(r, \sum_{s \in S} \mathbf{1}^\top \mathbf{z}_s), a)$ We set $r = \sum_{s \in V} \mathbf{1}^\top \mathbf{z}_s / 3$, $b = \sum_{s \in V} \mathbf{1}^\top \mathbf{z}_s / 6$, $a = \sum_{s \in V} \mathbf{1}^\top \mathbf{z}_s / 2$. We observe that our model significantly outperforms the baselines.

**Ablation study.** We compare different variants of our approach: (i) FLEXSUBNET, trained by decoupling the double integral into independent integrals (Eq. (14)); (ii) FLEXSUBNET, trained using end-to-end training via backpropagation through double integrals (Section 3.6); and (iii) FLEXSUBNET with $N = 1$, where $N$ is the number of steps of the recursions (1). Table 3 summarizes the results which reveal the following observations. (1) We achieve substantial performance gain via decoupling into independent integrals. Although decoupling integrals is an approximation of end-to-end training, it also provides a more smooth loss surface than the loss on the double integral. (2) Running FLEXSUBNET for only a single step significantly deteriorates performance.

**Performance with additional planted synthetic function.** Here, we consider a featurized form of the function used in the proof of the lower bound [31]:

$F(S) = \min(\sum_{s \in S} \mathbf{1}^\top \mathbf{z}_s, b + \min(r, \sum_{s \in S} \mathbf{1}^\top \mathbf{z}_s), a)$. We set $r = \sum_{s \in V} \mathbf{1}^\top \mathbf{z}_s/3$, $b = \sum_{s \in V} \mathbf{1}^\top \mathbf{z}_s/6$, $a = \sum_{s \in V} \mathbf{1}^\top \mathbf{z}_s/2$.

Table 4 summarizes the results in terms of RMSE. We observe that our method outperforms other methods by a substantial margin.

**Scalability analysis.** We report per-epoch training time of different methods in the context of training by (set, value) pairs.

| FLEXSUBNET | Deep-set | Set-transformer | DSF | SubMix |
|---|---|---|---|---|
| 7.2649 | 1.3009 | 2.2856 | 1.39 | 1.329 |

Table 5: Per epoch time (in second) for different methods.

While our method is slower than the baseline methods (mainly due to the numerical integration), it offers significantly higher accuracy than other methods.

**Variation of the performance of Set-transformer against the number of parameters.** Set-transformer is an excellent neural set function with very high expressive power. This expressive power comes from its ability to incorporate interaction between elements. However, it consumes significantly large memory in practice. Our method and all the other baselines do not incorporate interactions and thus consume lower memory. Thus, we reduced the number of parameters of Set-transformer so that it consumes similar memory as our method (8–10 GB) for a fair comparison. Here, we experiment with different batch size (B) and increased number (P) of parameters (as GPU memory permitted). Results are as follows.

| P=29, B=66 | P=139, B=40 | P=321, B=40 | P=321, B=17 | P=439, B=17 |
|---|---|---|---|---|
| 0.063 | 0.069 | 0.056 | 0.054 | 0.055 |

Table 6: RMSE for different configurations of Set transformer for Facility Location dataset

As expected, the performance indeed improves if we increase the number of parameters. However, even with significantly small batch size, an increased number of parameters (otherwise accuracy drops) led to large computation graphs with excessive GPU RAM consumption. This is because, in our problem, maximum set size $|V| = 10000$ and each instance in our problem is a featurized tensor of dimension $10 \times 10000$. We believe that processing batches of such instances leads to the set-attention-blocks consuming huge memory. As mentioned by Lee et al. [51, Page 16], the SAB block in Set-transformer admits a maximum size of 2000 elements, in contrast, we have 10000 elements.

# F    Additional details about experiments on subset selection for product recommendation

| Catgories | $|\mathcal{U}|$ | $|V|$ | $\sum |S|$ | $\mathbb{E}[|S|]$ | $\min_S |S|$ | $\max_S |S|$ |
|---|---|---|---|---|---|---|
| Gear | 4277 | 100 | 16288 | 3.808 | 3 | 10 |
| Bath | 3195 | 100 | 12147 | 3.802 | 3 | 11 |
| Health | 2995 | 62 | 11053 | 3.69 | 3 | 9 |
| Diaper | 6108 | 100 | 25333 | 4.148 | 3 | 15 |
| Toys | 2421 | 62 | 9924 | 4.099 | 3 | 14 |
| Bedding | 4524 | 100 | 17509 | 3.87 | 3 | 12 |
| Feeding | 8202 | 100 | 37901 | 4.621 | 3 | 23 |
| Apparel | 4675 | 100 | 21176 | 4.53 | 3 | 21 |
| Media | 1485 | 58 | 6723 | 4.527 | 3 | 19 |

Table 7: Amazon baby registry statistics.

## F.1    Dataset description

As mentioned in Section 5.2, each dataset contains a universal set $V$ and a set of subsets $\mathcal{S} = \{S\}$. We summarize the details of the categories of the Amazon baby registry [30] in Table 7. For each categories, we first filter out those subsets $S$ for which $|S| \geq 3$. Moreover, we use the 768 dimensional BERT embedding of the description of each item $s \in V$ to compute $\mathbf{z}_s$.

## F.2 Implementation details

We implemented FLEXSUBNET, DSF and SubMix following the procedure described in the Appendix D, except that we considered we use $F_\theta(S) = \theta_1 \log(1 + \sum_{s \in S} \theta_m^\top z_s) + \theta_2 \log(1 + \log(1 + \sum_{s \in S} \theta_m^\top z_s)) + \theta_3 \log(1 + (\log(1 + \log(1 + \sum_{s \in S} \theta_m^\top z_s))))$. for SubMix. Here, we train each trainable model for 30 epochs and choose the trained model that gives best mean Jaccard coefficient on the validation set in these 30 epochs. For maximizing DPP, Facility Location and Disparity Min baselines, we used the library https://github.com/decile-team/submodlib.

**Computing environment.** Our code was written in pytorch 1.7, running on a 16-core Intel(R) Xeon(R) Gold 6226R CPU@2.90GHz with 115 GB RAM, one Nvidia V100-32 GB GPU Card and Ubuntu 20.04 OS.

## F.3 License

We collected Amazon baby registry dataset from https://code.google.com/archive/p/em-for-dpps/ which comes under BSD license.

# G Additional experiments on real data

## G.1 Replication of Table 2 with standard deviation

Here, we reproduce Table 2 with standard deviation. Table 8 shows the results.

| | **Mean Jaccard Coefficient (MJC)** | | | | | |
|---|---|---|---|---|---|---|
| | FLEXSUBNET | DSF | SubMix | FL | DPP | DisMin |
| Gear | **0.101 ± 0.003** | 0.099 ± 0.003 | 0.028 ± 0.002 | 0.019 ± 0.001 | 0.014 ± 0.001 | 0.013 ± 0.001 |
| Bath | **0.091 ± 0.004** | 0.087 ± 0.003 | 0.038 ± 0.002 | 0.02 ± 0.002 | 0.012 ± 0.001 | 0.01 ± 0.001 |
| Health | **0.153 ± 0.005** | 0.142 ± 0.004 | 0.022 ± 0.002 | 0.084 ± 0.004 | 0.011 ± 0.001 | 0.015 ± 0.001 |
| Diaper | **0.134 ± 0.004** | 0.115 ± 0.004 | 0.023 ± 0.001 | 0.018 ± 0.001 | 0.013 ± 0.001 | 0.012 ± 0.001 |
| Toys | **0.157 ± 0.006** | 0.15 ± 0.006 | 0.025 ± 0.002 | 0.064 ± 0.003 | 0.029 ± 0.002 | 0.029 ± 0.002 |
| Bedding | **0.203 ± 0.005** | 0.191 ± 0.004 | 0.028 ± 0.002 | 0.015 ± 0.001 | 0.043 ± 0.002 | 0.047 ± 0.002 |
| Feeding | **0.1 ± 0.002** | 0.091 ± 0.002 | 0.026 ± 0.001 | 0.023 ± 0.001 | 0.02 ± 0.001 | 0.019 ± 0.001 |
| Apparel | **0.101 ± 0.003** | 0.093 ± 0.003 | 0.036 ± 0.002 | 0.022 ± 0.001 | 0.016 ± 0.001 | 0.016 ± 0.001 |
| Media | **0.135 ± 0.006** | 0.13 ± 0.006 | 0.029 ± 0.003 | 0.035 ± 0.003 | 0.029 ± 0.002 | 0.025 ± 0.002 |
| | **Mean Normalized Discounted Cumulative Gain@10 (Mean NDCG@10)** | | | | | |
| | FLEXSUBNET | DSF | SubMix | FL | DPP | DisMin |
| Gear | **0.539 ± 0.004** | 0.538 ± 0.004 | 0.449 ± 0.003 | 0.433 ± 0.002 | 0.425 ± 0.002 | 0.426 ± 0.002 |
| Bath | **0.52 ± 0.004** | 0.5 ± 0.004 | 0.447 ± 0.002 | 0.433 ± 0.002 | 0.427 ± 0.002 | 0.422 ± 0.002 |
| Health | **0.597 ± 0.005** | 0.549 ± 0.004 | 0.449 ± 0.003 | 0.54 ± 0.005 | 0.425 ± 0.002 | 0.435 ± 0.002 |
| Diaper | **0.562 ± 0.004** | 0.546 ± 0.004 | 0.447 ± 0.002 | 0.44 ± 0.002 | 0.435 ± 0.002 | 0.435 ± 0.002 |
| Toys | **0.591 ± 0.006** | 0.577 ± 0.006 | 0.446 ± 0.003 | 0.472 ± 0.003 | 0.448 ± 0.003 | 0.449 ± 0.003 |
| Bedding | **0.643 ± 0.005** | 0.623 ± 0.004 | 0.437 ± 0.002 | 0.438 ± 0.002 | 0.456 ± 0.002 | 0.461 ± 0.002 |
| Feeding | **0.55 ± 0.003** | 0.547 ± 0.003 | 0.459 ± 0.002 | 0.453 ± 0.001 | 0.454 ± 0.001 | 0.452 ± 0.001 |
| Apparel | **0.558 ± 0.004** | 0.55 ± 0.004 | 0.459 ± 0.002 | 0.452 ± 0.002 | 0.446 ± 0.002 | 0.444 ± 0.002 |
| Media | **0.578 ± 0.007** | 0.578 ± 0.006 | 0.474 ± 0.004 | 0.47 ± 0.004 | 0.461 ± 0.004 | 0.461 ± 0.004 |

Table 8: Replica of Table 2 with standard deviation. Here, we perform prediction of subsets in product recommendation task. Performance is measured in terms of Jaccard Coefficient (JC) and Normalized Discounted Cumulative Gain@10 (NDCG@10) for nine datasets the Amazon baby registry records, for FLEXSUBNET, Deep submodular function (DSF), mixture of submodular functions (SubMix), Facility location (FL), Determinantal point process (DPP) and Disparity Min (DisMin). In all experiments, we use training, test, validation folds of equal size. Numbers in **bold** (underline) indicate best (second best) performer.

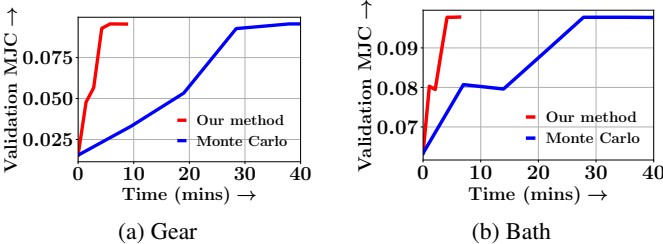

(a) Gear             (b) Bath

Figure 9: Variation of MJC as training progresses, for Gear and Bath categories. Our proposed permutation adversarial subset selection is atleast $4\times$ faster than the Monte Carlo sampling method [79].

## G.2  Efficiency

The key motivation of our proposed data subset selection method is to ensure that the estimated parameters are invariant to the order of the elements of the input subset. Tschiatschek et al. [79] achieve this goal by computing Monte Carlo average of the underlying likelihood over many samples. In Figure 9, we compare our method with their proposal, which shows that our method is $\geq 4\times$ faster.