# OpenReview forum: "Neural Estimation of Submodular Functions with Applications to Differentiable Subset Selection"
_NeurIPS.cc/2022/Conference — NeurIPS 2022 Accept_

### Official Review · Reviewer_z5K8 · 2022-07-10

**Rating:** 7
**Confidence:** 4
**Soundness:** 3 good
**Presentation:** 3 good
**Contribution:** 3 good

**Summary:**

The paper propose a family of neural models for monotone and non-monotone submodular functions and monotone approximately submodular functions, which can be estimated both from (set, value) observations and (perimeter-set, high-value subset) observations. The model proposed for monotone submodular functions consists of recursively applying an increasing concave activation function on a linear combination of the submodular function from the previous step and a positive modular function. This model is extended to a class of monotone approximately submodular functions by replacing the increasing concave activation function by an activation function satisfying a second-order differential inequality (which is stricter than concavity). The model proposed for non-monotone submodular functions consists of the composition of a concave activation function (not necessarily increasing) on only one positive modular function. Unlike prior work which use a fixed concave function, the authors choose to model the concave activation functions using a neural network which involve double integrals and propose methods to train it.  The authors also propose an order-invariant greedy sampling method for learning set functions from (perimeter-set, high-value subset) observations based on a max-min game between an adversarial permutation generator and the set function learner. Finally, they present some experiments showing that their models outperform other baselines in both supervision settings.

**Questions:**

- For a more fair comparison with the DSF baseline, it would be good to consider few other popular choices for the concave function.
- How are high-value subsets chosen in the Amazon baby registry dataset?

**Limitations:**

Authors discuss limitations of their work in Appendix A, but do not discuss potential negative societal impact.


**Strengths And Weaknesses:**


Strengths:
- Designing a practical method for learning submodular and approximately submodular functions is useful for a number of ML applications.
- Proposing a neural network model for the concave activation function which can be learned from data is a good alternative to having to choose a fixed one.
- Paper is well written with clearly presented results

Weaknesses:
- Unlike some prior work, the proposed models are only shown to be good estimators of monotone submodular function of the form of concave composed modular function, which is too limited.
- The proposed model for non-monotone submodular functions is too simple to capture complex functions.
- The empirical comparison with the deep submodular function (DSF) baseline is limited (see questions).
- Missing related work:

	Balcan, Maria-Florina, and Nicholas JA Harvey. Submodular functions: Learnability, structure, and optimization. SIAM Journal on Computing 47.3: 703-754, 2018.

	Vitaly Feldman and Jan Vondrak. Optimal bounds on approximation of submodular and xos functions by juntas. SIAM Journal on Computing, 45(3):1129–1170, 2016.

	R. Sipos, P. Shivaswamy, and T. Joachims. Large-margin learning of submodular summarization mod- els. In Proceedings of the 13th Conference of the European Chapter of the Association for Computational Linguistics, pages 224–233. Association for Computational Linguistics, 2012.

---

> ### Author Response · Authors · 2022-08-02
> **Response to Reviewer z5K8**
>
> Many thanks for your review, this would help improve our paper.
> > *Unlike some prior work, the proposed models are only shown to be good estimators of monotone submodular function of the form of concave composed modular function, which is too limited.*
>
> Flexsubnet models submodular functions that are of the form of concave composed submodular functions. We start with a concave composed modular function but then recursively construct submodular functions which are obtained by applying trainable concave functions on top of submodular functions computed in the previous step.
>
> In our experiments, we do consider graph cut functions which are  submodular functions but not concave composed modular functions. Moreover, during the rebuttal period, we also considered more challenging function, as suggested by Reviewer cuUV. To this end, we considered two functions:
>
> The first function is given by $F(S) = \min(\sum\_{s\in S} \pmb{1}^{\top} \pmb{z}\_s,  b+\min(r , \sum\_{s\in S}\pmb{1}^{\top} \pmb{z}\_s),a)$.
> We set $r = \sum\_{s\in V} \pmb{1}^{\top} \pmb{z}\_s / 3$ ,  $b =  \sum\_{s\in V} \pmb{1}^{\top} \pmb{z}\_s / 6$, $a =   \sum\_{s\in V} \pmb{1}^{\top} \pmb{z}\_s / 2$.
>
>
> The following table summarises the results in terms of RMSE for the above function.
>
> Our method | Set transformer | Deep set | DSF  | Submix
> ---------- | --------------- | -------- | ---- | ------
> 0.055      | 0.119           | 0.160    | 2.31 | 1.770
>
> The second function is given by:
> $F(S) = \min(\sum\_{s\in S} \pmb{1}^{\top} \pmb{z}\_s,  b+ \sum\_{s\in S\cap R}\pmb{1}^{\top} \pmb{z}\_s,a)$.
> We draw $R \subset V$ uniformly at random with |R| = |V|/3,  $b =  \sum\_{s\in V} \pmb{1}^{\top} \pmb{z}\_s / 6$, $a =   \sum\_{s\in V} \pmb{1}^{\top} \pmb{z}\_s / 2$.
>
> The following table summarises the results in terms of RMSE for the second variant:
>
> Our method | Set transformer | Deep set | DSF  | Submix
> ---------- | --------------- | -------- | ---- | ------
> 0.020      | 0.632           | 0.087    | 2.33 | 1.370
>
>  We observe that our method outperforms other methods by a substantial margin.
>
>
> > *The proposed model for non-monotone submodular functions is too simple to capture complex functions.*
>
> We agree. But in general, the non-monotone submodular functions are difficult to construct in a recursive manner.
>
>
> > *The empirical comparison with the deep submodular function (DSF) baseline is limited*
>
> We tried with two more concave functions, e.g.,  $\phi(x) = \sqrt x,  \phi(x) = a-\exp(-bx)$. We summarize the RMSE for facility location and Gcut$\_{0.1}$
> |                              | Facility location | Gcut$\\\_{0.1}$ |
> | ---------------------------- | ----------------- | --------------- |
> | Our method                   | 0.022             | 0.004           |
> | DSF (log) (already reported) | 0.189             | 0.778           |
> | DSF (sqrt)                   | 0.290             | 0.987           |
> | DSF (a-exp(-bx))             | 0.775             | 0.762           |
>
> We observe that DSF (log) performs significantly better for Facility location. Moreover, our method performs significantly better in Gcut$\_{0.1}$. Our analysis suggested that if the true concave function and the concave function hardcoded in DSF do not match each other, the performance of DSF can be very poor in many cases. Moreover, if we use more and more (wrong) concave functions recursively, the function becomes quickly flat for a slightly large size of S. Thus the training algorithm receives very weak gradient based signals.
>
> > *How are high-value subsets chosen in the Amazon baby registry dataset?*
>
> Amazon baby registry dataset comes directly with  a set of recommended subsets which are referred to as high-value subsets. For each category, we have two files. The first file contains <ID, category, description>. Following is an example of one line of the file.
> “4 diaper: Summer Infant Ultra Plush Change Pad Cover, Blue : Plush Animal Toys : Baby”
> The second file contains the list of recommended subsets in terms of the ids, e.g., one line of the file is “9,10,11.”
>
> > *Missing references*
>
>
> We added the references in the main paper.

---

> > ### Comment · Reviewer_z5K8 · 2022-08-08
> > **Response to authors**
> >
> > Thank you for addressing my questions and providing additional results. I recommend that you add them to the paper. I also second Reviewer cuUV's suggestion of adding a discussion about known theoretical results regarding sample complexity and running time of algorithms for learning/approximation submodular functions. The first two missing related work I mentioned in my review provide such results (see also references therein). I want to also clarify that in my comment
> >
> > > Unlike some prior work, the proposed models are only shown to be good estimators of monotone submodular function of the form of concave composed modular function, which is too limited.
> >
> > I was referring to the *theoretical results* you provide in Proposition 4, which only apply to concave composed with modular functions.

---

> > > ### Author Response · Authors · 2022-08-09
> > > **Response to reviewer's comments**
> > >
> > > Thank you for your suggestions.
> > > We have added a short discussion about sample complexity in Appendix B. lines 509–514. We plan to include more in the subsequent revision.  If our paper gets accepted, we will get space for one more page and will bring the discussion in main.
> > >
> > >
> > > We now better understand your concern about Proposition 4. Extending this proposition to a general class of submodular functions seems quite challenging and possibly hard to fit into the scope of a single paper. We leave it as future work.
> > >
> > > We believe FlexSubNet can (be adapted to) match the DSF class of Bilmes and Bai (2017), but, as Figure 4 in their paper shows, there is an unexplored region between DSFs and all submodular functions that might call for yet newer architectures.

---

### Official Review · Reviewer_9tyR · 2022-07-11

**Rating:** 6
**Confidence:** 4
**Soundness:** 3 good
**Presentation:** 3 good
**Contribution:** 3 good

**Summary:**

This paper proposes new families of submodular (and $\alpha$ approximate submodular) functions which can be parameterized efficiently by neural networks. Then the authors use this formulation to learn submodular functions given (set, value) examples. Additionally, they propose a new max-min method to learn to sample high-value subsets from a submodular function given (perimeter-set, high-value subset) examples. The inner min objective uses a Gumbel-Sinkhorn network to encourage permutation invariance.

**Questions:**

- How wide is each FlexSubNet layer? More generally, how many parameters does FlexSubNet have compared to baselines?
- The hyperparameter configuration looks very similar across most methods, how were these selected? If cross validation, what was the range of values for $N$, $\alpha$, number of hidden layers, etc.
- What values of $\alpha$ do ground truth planted set functions (v) and (vi) have?
- In Table 2, is N=1 with Decoupling or End-end?
- When using the Amazon baby registry dataset, how large are the sampled sets and prefixes?

**Limitations:**

The authors could have commented on the tradeoff between using (set, value) and (perimeter-set, high-value subset) datasets. The latter appears to have a higher risk of introducing bias into the learned model

**Strengths And Weaknesses:**

## Originality
- Novel combination of previous work (Deep Submodular Functions, Gumbel-Sinkhorn NNs, Unconstrained Monotonic NNs) with new insights and techniques.
- Cites previous related work where appropriate, except that Reference [42] in the Appendix does not appear in main paper

## Quality
- Several technical contributions: parameterizing (alpha) submodular functions efficiently, max-min formulation of selecting a greedy sequence via adversarial permutation
- Experimental results look impressive, but I have some concerns with experimental setup/evaluation of baseline methods:
    - For experiments involving with set transformer, using a batch size less than 66 could have reduced gpu memory consumption
    - On the Amazon baby registry dataset, can still use other baselines such as set transformer to predict the next element, even if submodularity is not enforced
    - The paper would be stronger if it evaluated performance in additional ways; for example, training time and sample complexity
    - Unclear why NDCG is an appropriate metric when it depends on order. The problem is designed to be permutation invariant

## Significance
- This paper presents interesting, potentially high-impact ideas for an important, relevant problem of learning submodular functions from data.
- In addition to the 2 planted $\alpha$-submodular set functions, the paper would be improved if it motivated this family of functions with practical applications

## Clarity
- This paper is organized well and is generally a pleasure to read
- While the theoretical details are quite clear, there are many missing details from the experimental evaluation. For example, no details on how the max-min objective is trained. Also it is somewhat unclear how the product recommendation dataset is converted into a (perimeter set, high-value subset) format.
- Note that some references such as [1] use $1 - \alpha$ in the definition of approximate submodularity.


[1]  Gatmiry and Gomez Rodriguez. Non-submodular Function Maximization subject to a Matroid Constraint, with Applications. https://arxiv.org/abs/1811.07863

---

> ### Author Response · Authors · 2022-08-02
> **Response to Reviewer 9tyR (Part - 1/2)**
>
> Many thanks for your reviews. They would help us improve our paper.
>
> >  *set transformer: using a batch size less than 66 could have reduced gpu memory*
>
> We experimented with lower batch size and increased number of parameters (as GPU memory permitted) of set transformer.  When we increased the number of parameters (P), we had to reduce the batch size  (B) to fit it within available GPU memory.  We observed some benefits for a long sweep of (P,B) in this context. Below are the RMSE numbers for Facility Location dataset for different (P, B) pairs.
>
>  P = 29, B =66|P = 139, B = 40|P = 321, B = 25|P = 321, B =17|P = 439, B =17
> -------------|---------------|---------------|--------------|--------------
> 0.067        |0.069          |0.056          |0.054         |0.055
>
> Even after such wide sweep of no. of parameters and batch size, our method (P=423, B=66, RMSE = 0.022) outperforms set transformer.  Note that our method consumes significantly lower GPU memory (12 GB at max), whereas the best variant of set transformer (# parameters = 321, batch size = 17)  consumes 23.1 GB  GPU memory.  Thus, even with significantly small batch size, an increased number of parameters (otherwise accuracy drops) leads to high GPU consumption. This is because, in our problem, maximum set size |V| = 10000 and each instance in our problem is a featurized tensor of dimension 10 x 10000. We believe that processing batches of such instances leads to the set-attention-blocks consuming huge memory, as the set transformer paper itself also provides such indications — the last sentence of page 16 of their arxiv version says that:
>
> “The maximum set size we report for SAB is 2,000 because the computation graph of bigger sets could not fit on our GPU.” (Here, GPU=Tesla P40 which is a 24GB GPU).
>
> In our case, the max set size is 10000, which made  set transformer training very difficult.
>
> > *On the Amazon baby registry dataset, can still use other baselines [...] even if submodularity is not enforced*
>
> Deep sets or set transformers guarantee neither submodularity nor monotonicity — the marginal gain $F(e | S) = F(S \cup e)- F(S)$ need  not be positive. Therefore, a greedy algorithm may be severely misled by such non-monotone models as deep set or set transformer. Consequently, the probabilistic greedy model, which is a differentiable trainable mechanism for greedy heuristics, may become quite ineffective.
>
> Combinatorial optimization of general non-monotone set functions is quite challenging, let alone designing a differentiable neural network for this task. There are approaches [1] to adapt greedy selection to non-monotone F if it can be expressed as F(S) = G(S) - m(S), where G is submodular and m is modular. With the help of this, we could tackle the problem of applying greedy algorithm on deep set/ set transformers using three steps: (i) We express the above non-monotone set function F(S) as a difference between two monotone submodular functions (F(S) = G(S) - H(S)). (ii) We approximate H(S) to a modular function $m\_H(S)$ which would lead to $F(S) \approx G(S)-m\_H(S)$. (iii) This can be maximized using distorted greedy algorithm [1].
>
> [1] Harshaw et al. "Submodular maximization beyond non-negativity. Guarantees, fast algorithms, and applications." ICML 19.
>
> However, given a function $F(S)$, it is extremely challenging to find out monotone submodular functions $G(S)$ and $H(S)$ in the first step (i) — this makes the probabilistic formulation for the above pipeline (i-iii) difficult for a set transformer or deep set.
>
>
> > *Unclear why NDCG is an appropriate metric when it depends on order. The problem is designed to be permutation invariant*
>
> While the greedy algorithm outputs the final set S, it also induces an order in which elements are chosen. The element $e$ that gives the highest value of $F(e)$, is chosen first; the element $e’$ which gives the highest value of $F(e’ \cup e)-F(e’)$ is chosen next and so on. Moreover, we expect that the marginal gain will decrease as we choose more and more elements. Thus, it is intuitive that the elements that provide more utility are chosen first. NDCG captures how well the learned set function captures the relative utility across elements.
>
> > *Training time comparison*
>
> We did compare the training time of our adversarial training method for predicting high value subset with Tschiatschek et al. which adopts a Monte Carlo approach to ensure permutation invariant training. Figure 4 in the main paper summarizes the results which shows that our method is 4x times faster. We further report per-epoch training time of different methods in the context of training by (set, value) pairs.
>
> Ours | Deep Set | Set transformer | DSF  | SubMix
> ---------- | -------- | --------------- | ---- | ------
> 7.2649     | 1.3009   | 2.2856          | 1.39 | 1.329
>
> While our method is slower than the baselines (due to the numerical integration), it offers significantly higher accuracy than other methods.

---

> > ### Author Response · Authors · 2022-08-02
> > **Response to Reviewer 9tyR (Part - 2/2)**
> >
> > > *For example, no details on how the max-min objective is trained.*
> >
> >  As argued in L283–287, we feed the feature matrix $\pmb{Z}\_S$ into an additional neural network $G\_{\omega}$ which is shared across different $(V,S)$ pairs and feed it into the Gumbel Sinkhorn network described in Eq. (20).  Then the above optimization reduces to $\max\_{\theta} \min\_{\omega}\sum\_{(V,S)\in U}\log \text{Pr}{}\_\theta (\omega, S)$, with $\omega$ taking the place of $\pmb{P}$. Now this is a continuous max-min problem wr.t. To $\theta$ and $\omega$. Likewise usual max-min training we follow the following procedure:
> >
> >
> >
> >     For each epoch e:
> >
> >               For each pair of $(V,S)\in U$:
> >
> >                    Fix $\omega$ and make gradient ascent update on $\theta$
> >
> > 	               Fix $\theta$ and make gradient descent update on $\omega$
> >
> > >  *how the product recommendation dataset is converted into a (perimeter set, high-value subset) format.*
> >
> >
> > Amazon baby registry dataset comes directly with a set of all items (perimeter set) and a set of recommended subsets (high-value subsets). The first file contains <ID, category, description>. Following is an example of one line of the file.
> >
> > “4 diaper: Summer Infant Ultra Plush Change Pad Cover, Blue : Plush Animal Toys : Baby”
> >
> > The second file contains the list of recommended subsets in terms of the ids, e.g., one line of the file is “9,10,11.”
> >
> > > *how many parameters does FlexSubNet have compared to baselines?*
> >
> > Our model: For end-to-end training, our method has 263 parameters and for decoupled training, our model has 423 parameters. For the latter, we need to introduce another neural network $\phi’\_{\theta}$ in Eq (13) for decoupling, which renders the total number of parameters to be 423.
> >
> > Deep set: The total number of parameters of the deep set is 284.  The deep set model is given by  $F(\sum\_{i\in S} m(x\_i))$. Here, we kept the architecture of $F$ exactly the same as the integrand $h\_{\theta}$ in our method. During the rebuttal period, we conducted experiments where the number of parameters (P) of a deep set is allowed to exceed our model size (P = 423). The following table reports RMSE numbers for facility location dataset.
> >
> >  Deep Set params $\\to$ | P = 534 | P = 1034 | P = 2034
> > ---------------------- | ------- | -------- | --------
> > RMSE                   | 0.148   | 0.153    | 0.175
> >
> > We observe that for even such an increased number of DeepSet parameters, our method outperforms it (RMSE = 0.022, # of parameters 423).
> >
> > Set transformer: Set transformer contains 29 parameters with batch size = 66.   Unfortunately, with the same batch size, we could not load it with more parameters in our GPU. We experimented with lower batch size (B) and increased number (P) of parameters  (as GPU memory permitted).  For more details, please refer to the response to your query *"For experiments involving with set transformer, using a batch size less than 66 could have reduced gpu memory consumption"* in the first part.
> >
> > DSF: 21 parameters, which is set using cross validation. We attempted to train DSFs for various model sizes by varying the number of recursive layers (we attempted {2,5,10,30,40} layers). We observe that it works best for 2 layers (this is also the case in our method where $N=2$). With an increased number of layers (max layers= 40 corresponds to number of parameters = 401), we observe that the performance does not provide any improvement. This is because, when we apply many concave functions one after another, the gradient often quickly becomes very small for a slightly large size of $S$.
> >
> > Submix: We use linear combinations of  four submodular functions. Hence, the number of parameters is 4.
> >
> > We added this  discussion in Appendix E.4.
> > > *Hyperparameters: the range of values for N, alpha ,hidden layers, number of hidden layers, etc.*
> >
> > Range of $N$ is {1,2,3,4} and we found the best validation set error at $N=2$. The search space of $alpha$ is [0.1,0.3,0.5.,0.7,0.8,0.9]. We did not search explicitly for the number of hidden layers of any of the models. During implementation, we chose model size so that each of the most heavy models, i.e., our model, deep set and set transformer, fits well into a moderate size GPU (12–16GB GPU). We could not do it with a set transformer with a batch size 66 and 29 parameters and we resorted to a  larger GPU (24GB and 32GB).
> >
> > > *alpha of ground truth planted set functions*
> >
> > We have provided a theoretical lower bound on alpha in Appendix E.5. We did not use these lower bounds while learning our model and rather used them as hyperparameters. In all cases, we found $\alpha \in (0.8,0.9)$.
> >
> > In Table 2, we use decoupling for N=1 and for sampled sets and prefixes for amazon dataset, please refer to Table 7 in appendix.

---

> > > ### Comment · Reviewer_9tyR · 2022-08-08
> > > **Follow up**
> > >
> > > Thank you for providing additional experiments for set transformers with different batch sizes, and for answering my questions about training parameters, training time, hyperparameters, and max-min optimization via gradient ascent-decent

---

### Official Review · Reviewer_cuUV · 2022-07-14

**Rating:** 6
**Confidence:** 5
**Soundness:** 3 good
**Presentation:** 3 good
**Contribution:** 3 good

**Summary:**

The paper considers the problem of learning submodular functions using neural networks. More precisely the paper considers two different settings.

1 - There exists an oracle that given a set $S$, responds with the value of the function $f$ for $S$.

2- There exists an oracle function that given a set $S$, returns $\argmax_{T\subseteq S} f(T)$.

The first setting is known as the value oracle in the literature and the second one can be simulated by demand oracle (put the price of infinity on any item not in set $S$). For the first setting, the goal is to minimize $\sum_{i \in [I]} (f(S) - F_\theta(S))$ over the parameters $\theta$ of a neural network. The paper proposes a specific class of submodular functions that arise from concave combination of modular functions in a recursive way and a neural network based on this class of functions. They deal with some issues for learning these functions such as permutation invariance in the input sets and dealing with the non-positivity of the second-order derivative. They also show that their approach can be extended to the class of $\alpha$-submodular functions and non-monotone submodular functions. The experiments also illustrates that on some synthetic data and a real-world dataset (Amazon baby registry dataset), they outperform other neural networks such as transformers, deep submodular functions, and deep sets.

**Questions:**

- The characterization of \alpha-submodular functions described in Theorem 2 is very similar to the recently introduced notions of one-sided smoothness and meta-submodularity. Please see [2] and [3]. These notions also consider functions where the second-order derivative is bounded by some factor of the first-order derivative. Please add a discussion about the connections. Is this a special case of one-sided smoothness? I'm asking because the regularization with the size of the set is looser than $1/k$.

- What is exactly the objective function for the second setting with (perimeter-set, high-value-subset) setting. Is it some ordering function? Please add a discussion about this. I don't think it is clear in the paper.

Comments:

- I think it is worth mentioning in the paper that learning submodular functions with poly(n) number of value oracle queries is hard in the worst case. Please see [1].

- Moreover, some discussion on the difference of theoretical upper bounds and lower bounds of approximation for the two different setting considered in the paper is useful, i.e., value-oracle setting and demand oracle setting.

- In Line 84, “and” is repeated.

References:

[1] Goemans, Michel X., Nicholas JA Harvey, Satoru Iwata, and Vahab Mirrokni. "Approximating submodular functions everywhere." In Proceedings of the twentieth annual ACM-SIAM symposium on Discrete algorithms, pp. 535-544. Society for Industrial and Applied Mathematics, 2009.

[2] Ghadiri, Mehrdad, Richard Santiago, and Bruce Shepherd. "Beyond submodular maximization via one-sided smoothness." In Proceedings of the Thirty-Second Annual ACM-SIAM Symposium on Discrete Algorithms, pp. 1006-1025. 2021.

[3] Ghadiri, Mehrdad, Richard Santiago, and Bruce Shepherd. "A Parameterized Family of Meta-Submodular Functions." arXiv preprint arXiv:2006.13754 (2020).


**Limitations:**

The paper does not have any negative societal impact since it considers a fundamental problem.


**Strengths And Weaknesses:**

The paper considers an interesting and fundamental problem. The paper is mostly written clearly and cleanly, but some discussions regarding the below topics could help understand its position in the literature better.

I think the paper makes a good case regarding its approach, but I'm not sure if some of the empirical comparisons make sense. For example, my understanding is that the deep sets essentially suggest a layer of a neural network that guarantees invariance/equivariance for datasets comprising of sets. This does not give a specific neural network but rather can be used as a component of more complicated networks. I wonder how the suggested method would interact with such a layer to guarantee order invariance. Please let me know if there is an issue that prevents this combination. Another concern is that all of the considered functions are simple concave/submodular functions. I'm wondering how these methods would compare on more extreme examples. For example, one candidate could be the worst-case example mentioned in [1] to prove the lower bound for approximation submodular functions.

Moreover, some discussion on the learnability of the functions could improve the paper. For example, in Proposition 4, it is mentioned that there exists a neural network with a certain width to learn a certain submodular function, but there is no mention of the number of samples needed or the running time.

---

> ### Author Response · Authors · 2022-08-02
> **Response to the concerns of Reviewer cuUV (Part-1/2)**
>
> Many thanks for your feedback, which would improve our paper. We added some of the discussions including new experiments and the discussion about one-sided-smoothness  in Appendix. If our paper gets accepted, we will add them in the extra page allowed in final revision. Comment on hardness is given in main.
>
> > *Deep sets [..] guarantees invariance/equivariance for datasets comprising of sets. This does not give a specific neural network but rather can be used as a component of more complicated networks. I wonder how the suggested method would interact with such a layer to guarantee order invariance. Please let me know if there is an issue that prevents this combination.*
>
> There are two places where element order (-invariance) is important — (a) the design of F(S), and (b) the selection of high-value subsets using a greedy method with f(S) (which is assumed to be order-invariant). We discuss below the possibilities you suggest wrt both (a) and (b).
>
> (a) Design of F(S): We have ensured the exact order-invariance of F(S) trivially by starting with modular functions of elements. In Table 1,  we have already compared FlexSubNet against a baseline F(S) implementation where S is input to a Deep Set network, followed by a MLP to return a value of S.
>
> (b) High value subset selection: Here, we have a probabilistic greedy model which takes the elements of a high value subset  $S$ in some order ($\pi$) as input. Thus, the likelihood of each element being selected depends on all the elements inserted before. Please refer to Eq. 16— the likelihood is on the sequence input. This renders the estimated parameters to be dependent on $\pi$ even though the set function $F\_{\theta}(S)$ is permutation invariant. The goal of our adversarial training is to ensure permutation invariance of the learned model parameters rather than the set function itself (since it is already permutation invariant).
>
> Deep sets or set transformers guarantee neither submodularity nor monotonicity — the marginal gain $F(e | S) = F(S \cup e)- F(S)$ need  not be positive. Therefore, a greedy algorithm may be severely misled by such non-monotone models as deep set or set transformer. Consequently, the probabilistic greedy model, which is a differentiable trainable mechanism for greedy heuristics, would be quite ineffective. For more details, please refer to the response to "*On the Amazon baby registry dataset, can still use other baselines such as set transformer to predict the next element, even if submodularity is not enforced*" in the response to Reviewer 9tyR.
>
> > *Another concern is that all of the considered functions are simple concave/submodular functions. I'm wondering how these methods would compare on more extreme examples. For example, one candidate could be the worst-case example mentioned in [1] to prove the lower bound for approximation submodular functions.*
>
> Note that we also considered graph cut function which cannot expressed using simple concave functions. If we understand correctly by the type of function used in the proof of lower bound [1], it is of the form of $\min (|S|, \beta + |S\cap R|, \alpha)$. We consider two similar featurized set functions.
>
>
> The first one is given by $F(S) = \min(\sum\_{s\in S} \pmb{1}^{\top} \pmb{z}\_s,  b+\min(r , \sum\_{s\in S}\pmb{1}^{\top} \pmb{z}\_s),a)$.
> We set $r = \sum\_{s\in V} \pmb{1}^{\top} \pmb{z}\_s / 3$ ,  $b =  \sum\_{s\in V} \pmb{1}^{\top} \pmb{z}\_s / 6$, $a =   \sum\_{s\in V} \pmb{1}^{\top} \pmb{z}\_s / 2$.
>
>
> The following table summarises the results in terms of RMSE for the above variant.
>
> Our method | Set transformer | Deep set | DSF  | Submix
> ---------- | --------------- | -------- | ---- | ------
> 0.055      | 0.119           | 0.160    | 2.31 | 1.770
>
> The second variant is given by:
> $F(S) = \min(\sum\_{s\in S} \pmb{1}^{\top} \pmb{z}\_s,  b+ \sum\_{s\in S\cap R}\pmb{1}^{\top} \pmb{z}\_s,a)$.
> We draw $R \subset V$ uniformly at random with |R| = |V|/3,  $b =  \sum\_{s\in V} \pmb{1}^{\top} \pmb{z}\_s / 6$, $a =   \sum\_{s\in V} \pmb{1}^{\top} \pmb{z}\_s / 2$.
>
> The following table summarises the results in terms of RMSE for the second variant:
>
> Our method | Set transformer | Deep set | DSF  | Submix
> ---------- | --------------- | -------- | ---- | ------
> 0.020      | 0.632           | 0.087    | 2.33 | 1.370
>
>  We observe that our method outperforms other methods by a substantial margin.
> Note that, our model first starts with concave on modular function and then again apply concave function on this concave composed modular function. The above function also admits such similar structure and therefore, our model can characterize it well.

---

> > ### Author Response · Authors · 2022-08-02
> > **Response to the concerns of Reviewer cuUV (Part-2/2)**
> >
> > > *but there is no mention of the number of samples needed or the running time.*
> >
> > We did compare the training time of our adversarial training method for predicting high value subset with Tschiatschek et al. which adopts a Monte Carlo approach to ensure permutation invariant training. Figure 4 in the main paper summarizes the results, which show that our method is 4x faster.  In the context of training by (set, value) pairs, our method takes ~7.26 seconds for one epoch (with batch size=66) in a machine with Titan RTX 24GB GPU with 32 CPU cores and 500 GB RAM.
> >
> > Theoretical sample complexity analysis for our model is extremely challenging— the presence of a double integral makes it immensely difficult. In our experiments, we used 3333 training instances to train our model from (set, value) pairs (L312-L313). The details of the number of instances in the Amazon dataset used in learning from (perimeter set, high-value-subset) pairs is given in Table 7.
> >
> > > *Theorem 2 is very similar to the recently introduced notions of one-sided smoothness and meta-submodularity. …. [2] and [3] consider functions where the second-order derivative is bounded by some factor of the first-order derivative. Please add a discussion about the connections. Is this a special case of one-sided smoothness?*
> >
> >
> > We thank the reviewer for these references.  We also think that Theorem 2 describes a special case of one-sided smoothness (OSS). However, the significance of these characterizations are different between their and our work. First, they consider $\gamma$-meta submodular function which is a  different generalisation of submodular functions compared to $\alpha$-submodular functions. Second, the OSS characterization they provide is for the multilinear extension of $\gamma$-meta submodular function, whereas we provide the characterization of $\alpha$-submodular functions itself, which allows direct construction of our neural models. We did think about using multilinear extensions of submodular functions to design its neural model. However, we found the key bottleneck is that it is challenging to reconstruct the set function from the multilinear extensions.
> >
> > We added a discussion in Appendix B about this.
> >
> >
> > > *What is exactly the objective function for the second setting with (perimeter-set, high-value-subset) setting. Is it some ordering function? Please add a discussion about this. I don't think it is clear in the paper.*
> >
> >
> > For a given sequence $\pi=( s_1,.., s_n ) $ of elements in the set $S$, the likelihood function is given by
> >
> >
> > $ \text{Pr} (\pi;\theta) =\prod\_{j=1}^{k-1} \frac{\exp(\tau F (s\_{j+1} |  S\_j))}{\sum\_{s\in V \backslash S\_j}\exp(\tau F (s\_{j+1} | S\_j))}$.
> >
> >
> > Now, if we maximize $ \text{Pr} (\pi;\theta) $ wrt $\theta$, the trained parameter $\theta$ will depend on the permutation $\pi$. To overcome this problem, our goal is to estimate the parameters $\theta$ that would maximize the minimum likelihood induced by the worst case permutation $\pi\_{\min}$. Thus, in terms of the permutation \emph{matrix} $P$, our training objective becomes:
> >
> >
> > $\max\_{\theta} \min\_{P} \textstyle\sum\_{(V,S)\in U} \log \text{Pr}\_\theta (P, S)$
> >
> >
> > However, searching over possible permutation matrices $P$ can be daunting. Therefore, we resort to the Sinkhorn network which generates doubly stochastic matrices, as a surrogate of permutation matrices in a differentiable manner (Eq. 20). Here, we start with a seed matrix $B\_S$ which is computed by applying  a neural network $G\_{\omega}$ (Linear-ReLU-Linear) on the feature matrix  $Z\_S$ and then recursively apply Eq. 20 on top of it. Here, $B\_S =  G\_{\omega}(Z\_S)$.
> > Hence, the final training problem becomes:
> >
> >
> > $\max\_{\theta} \min\_{\omega} \textstyle\sum\_{(V,S)\in U} \log \text{Pr}\_\theta ( \text{Sinkhorn}(G\_{\omega}(Z\_S)), S)$.
> >
> >
> >
> > > *Moreover, some discussion on the difference of theoretical upper bounds and lower bounds of approximation for the two different settings considered in the paper is useful, i.e., value-oracle setting and demand oracle setting.*
> >
> >
> > Theoretical bounds for sample complexities learning from (set, value) pairs and from (perimeter, high-value-subset) pairs are extremely difficult in our context, given the complex neural form of the underlying functions and therefore we leave it as a future work.

---

> > > ### Comment · Reviewer_cuUV · 2022-08-07
> > > **Updating review**
> > >
> > > I'd like to thank the authors for the detailed response to my comments. Here are some minor comments about the response.
> > >
> > > 1. Although one-sided smoothness is used in [3] as a way of characterizing meta-submodularity, it was first introduced separately in [2] for any monotone twice-differentiable continuous function on any number of dimensions (even one dimension as your characterization of $\alpha$-submodular functions) and over subdomains of functions. It is very interesting that such a connection exists between these two different generalizations of submodular functions: meta-submodularity and $\alpha$-submodular functions.
> > >
> > > 2. I think it would be valuable to mention the theoretical sample complexity analysis of your work as future works in the conclusion. Moreover, it is valuable to add a discussion about _known_ theoretical results regarding sample complexity and running time of algorithms for learning/approximation submodular functions.
> > >
> > > The authors have addressed most of my concerns, and so I'd like to increase my score for the paper.

---

> > > > ### Author Response · Authors · 2022-08-09
> > > > **Response to the minor comments**
> > > >
> > > > Thank you for your suggestions. We have added a short discussion about sample complexity in Appendix B. lines 509–514. We plan to include more in the subsequent revision. If our paper gets accepted, we will get space for one more page and will bring the discussion in main.

---

### Meta-Review · Area_Chair_BmNx · 2022-08-24

**Recommendation:** Accept
**Confidence:** Certain

**Metareview:**

All 3 knowledgable reviewers recommended acceptance of the paper and had active discussions with the authors. I agree, that the paper makes several interesting and relevant contributions and suggest acceptance of the paper. Please consider the reviewers' comments when preparing the final version of the paper.

**Award:**

No

---

### Decision · Program_Chairs · 2022-09-14

Accept